# Spatio-temporal analysis of prostate tumors in situ suggests pre-existence of treatment-resistant clones

Maja Marklund [1,6], Niklas Schultz[2,6], Stefanie Friedrich [3,6], Emelie Berglund [1], Firas Tarish[2], Anna Tanoglidi[4], Yao Liu [2], Ludvig Bergenstråhle [1], Andrew Erickson[5], Thomas Helleday [2], Alastair D. Lamb [5], Erik Sonnhammer [3] ✉ & Joakim Lundeberg [1] ✉

The molecular mechanisms underlying lethal castration-resistant prostate cancer remain poorly understood, with intratumoral heterogeneity a likely contributing factor. To examine the temporal aspects of resistance, we analyze tumor heterogeneity in needle biopsies collected before and after treatment with androgen deprivation therapy. By doing so, we are able to couple clinical responsiveness and morphological information such as Gleason score to transcriptome-wide data. Our data-driven analysis of transcriptomes identifies several distinct intratumoral cell populations, characterized by their unique gene expression profiles. Certain cell populations present before treatment exhibit gene expression profiles that match those of resistant tumor cell clusters, present after treatment. We confirm that these clusters are resistant by the localization of active androgen receptors to the nuclei in cancer cells post-treatment. Our data also demonstrates that most stromal cells adjacent to resistant clusters do not express the androgen receptor, and we identify differentially expressed genes for these cells. Altogether, this study shows the potential to increase the power in predicting resistant tumors.

The clinical behavior of prostate cancer (PCa) is diverse; most tumors display slow and gradual growth, but for a smaller subset the development profile is more aggressive, and once spread outside the prostate it is ultimately a lethal disease. Most men with non-organ-defined PCa receive androgen deprivation therapy (ADT), which is initially successful but inevitably transforms into a non-responsive, lethal form within a few years, called castration-resistant PCa (CRPC)[1–5].

Primary PCa is often multifocal and consists of a dominant genetic clone accompanied by several less prevalent ones. Tumor clones have been observed to carry different genetic alterations, which implies the presence of spatial heterogeneity[6,7]. Due to the diverse molecular landscape of each individual prostate tumor, it has proven challenging to establish a reliable strategy for risk stratification and prediction of treatment outcome. Hence, to increase the treatment efficiency and improve chances of patient survival, it is essential to properly understand the biology behind this heterogeneity, and the molecular pathways that cause CRPC.

Currently, to construct a treatment plan, the morphology-based Gleason grading system is used as a first classification of prostate tumors, applied on a set of needle biopsies of the prostate obtained under ultrasound guidance. This system stratifies PCa into different Gleason Scores, ranging from 5 to 10, which further is divided into a

[1]Department of Gene Technology, KTH Royal Institute of Technology, Science for Life Laboratory, Solna, Sweden. [2]Division of Translational Medicine & Chemical Biology, Karolinska Institute, Science for Life Laboratory, Solna, Sweden. [3]Department of Biochemistry and Biophysics, Stockholm University, Science for Laboratory, Solna, Sweden. [4]Department of Pathology, Evangelismos General Hospital, 45-47 Ipsilantou str, Athens, Greece. [5]Nuffield Department of Surgical Sciences, University of Oxford, Oxford, UK. [6]These authors contributed equally: Maja Marklund, Niklas Schultz, Stefanie Friedrich. ✉e-mail: erik.sonnhammer@scilifelab.se; joakim.lundeberg@scilifelab.se

simpler form called Gleason Grade Groups (GGs, ISUP 2014), ranging from one to five, where a higher score is associated with a worse outcome in both cases[8]. While the Gleason grading system remains the gold standard, it is not perfect, with patients assigned to the same GG experiencing different disease progression and outcomes[9]. To enhance this classification, additional complementary analysis based on molecular methods have been considered, with a view to establishing a more robust and accurate characterization of PCa[10–14]. Currently, few PCa prognoses and treatments are based on tumor gene expression.

The androgen receptor (AR) plays a critical role for the survival and proliferation of prostate cancer cells. Under normal conditions, androgens bind to the AR, which translocates from the cytoplasm to the nucleus to function as a transcription factor, activating a cascade of essential genes required for prostate growth[15–21]. Therefore, AR-activity can be assessed by its localisation to the nucleus.

In CRPC, the normal dependence on androgens for survival and proliferation is bypassed[22]. Several studies have been conducted which describe the molecular basis and mechanisms of androgen deprivation and CRPC[17,23–26].

For decades, two models have been used to explain the origin of CRPC; adaptation and selection. The adaptation model refers to ADT-induced long-term shortage of androgens, resulting in an evolutionary pressure on tumor cells and the microenvironment, which favors changes that make tumor cells independent of androgen and thus promotes the development of CRPC. The selection model describes the pre-existence of rare, castration-resistant cells surviving during ADT and outgrowing the other cells that disappear due to induced apoptosis under the selective pressure of a low-androgen-environment[27–35]. It is likely that, regardless of which model predominates, selective targeting of these adaptive or pre-existing cells could prove a clinically useful strategy[36,37].

A limited set of studies have compared the time from ADT onset to ADT resistance as a clinical endpoint, with median duration ranging from 10–23.7 months. However, it has been shown that patients with more advanced disease develop CRPC at a faster pace after ADT-onset, which might indicate a greater proportion of androgen independent cells before treatment[38,39]. Furthermore, a lower prostate specific antigen (PSA) nadir (lowest measured PSA value after treatment onset) correlates with a longer interval before the development of CRPC[39–42].

Prostate tissue is composed of epithelial cells surrounded by stroma. The stromal cells in adults are mainly composed of smooth muscle cells and fibroblasts, but also of nerves, blood vessels, immune and inflammatory cells. During carcinogenesis the stromal compartment changes and is characterized by a major loss of smooth muscle cells and an increase in myofibroblasts and collagen fibers[43–45]. Evidence suggests that the tumor microenvironment plays a key role in malignant progression[46–49].

Spatial Transcriptomics[50] (ST) overcomes many of the limitations with non-spatial RNA sequencing methods such as bulk and single cell analysis. The ST technology links transcriptome-wide profiling with tissue morphology by implementing a barcoding scheme on the surface of a glass slide[50,51]. This technology is valuable for studies of the tumor and its microenvironment, since information regarding the spatial position of each observed transcript is preserved.

We use a model-based probabilistic framework, Spatial Transcriptome Decomposition[52] (STD), to perform a data-driven analysis of the gene expression data, which allows for hidden mixture interpretation represented by the non-homogenous cell type composition of the spots. In brief, STD decomposes the spatial gene expression into patterns across the tissue sections, referred to as factors, each representing a distinct gene expression profile with a corresponding spatial activity map. Each factor can be approximated to a specific cell type, cell state, microenvironment, or tissue component, representing different histological conditions. Then, a dimensionality reduction of the

preferred choice can be done, e.g. UMAP or tSNE, to visualize the full repertoire of factors in a single image. The full model is described elsewhere[53].

In this study, we identify unique gene expression profiles for ADT responding and non-responding tumors, irrespective of Gleason scores. Furthermore, we define a stromal gene expression profile that appears mainly to be present in non-responding patients. Our analysis opens up the potential to identify high-risk patients at the time of diagnosis, but also to understand the interplay between tumor and stroma providing clues to prevent resistance.

## Results

### Heterogeneous response of prostate tumor cells to castration

We have previously demonstrated that ADT overall reduces the expression of non-homologous end joining repair proteins, such as Ku70 and P-DNA-PKcs, and thereby hampers the DNA repair capacity of PCa cells, which agrees with increased response to radiation therapy[54]. However, we noticed an extensive variability in the levels of these repair proteins in the biopsies, both within and between patients.

To further investigate this, we analyzed a set of formalin-fixed needle biopsies pre- and post-ADT from five patients with Gleason scores ranging from six to nine (GG 1–5) and analyzed repair protein expression patterns along with nuclear AR, using immunohistochemistry (IHC). Epithelial nuclei were distinguished by an in-house-developed software (Fig. 1a), and epithelial nuclear AR intensities were measured from biopsies pre- and post-ADT, showing the presence of nuclear AR also post-ADT in a fraction of the epithelial glands, in all five patients, which indicates treatment unresponsiveness (Fig. 1b, c).

We found evidence that high levels of nuclear AR predict high levels of said repair proteins and a correlation between levels of nuclear AR and serum PSA post-ADT (Supplementary Fig. 1; Supplementary Table 1). Non-responsiveness, was thus strongly linked to the expression of nuclear AR post-ADT. ADT resistance may also be the result of the activation of other genes or oncogenic pathways. To gain a broader molecular understanding of implicated genes and pathways contributing to ADT resistance in situ we continued with an exploratory transcriptome analysis.

### Design of spatial gene expression experiments

To chart the molecular landscape of non-responsive cell clusters, fresh core needle biopsies pre- and post-ADT were collected from the prostates of three patients. The biopsies were snap frozen to facilitate gene expression analysis. We investigated the spatial gene expression profiles of each biopsy using ST[50], a transcriptome-wide methodology capturing mRNA from tissue sections using spatially barcoded spots on microscopic slides, followed by a data driven analysis that identifies gene expression patterns across the tissue sections (overviewed in Supplementary Fig. 2).

In total eight core needle biopsies per patient were analyzed; four biopsies pre-ADT and four biopsies eight weeks after gonadotropin-releasing hormone (GnRH)-analogue treatment (post-ADT). Clinical data during this period were collected (Supplementary Table 2). Sections from each biopsy were hematoxylin and eosin (H&E) stained, scanned, independently annotated by two pathologists, and analyzed with ST. Adjacent tissue sections were immunostained for AR to enable a comparison to in situ gene expression patterns. Spatially variable high levels of nuclear AR were observed in the new set of fresh frozen biopsies, validating previous observations (exemplified in Fig. 2a).

The histological annotations of the tissue sections were conducted using the Gleason grading system. Information regarding immune response/inflammation and high-grade prostatic intraepithelial neoplasia (HGPIN) was also notated (Supplementary Fig. 3). In addition, the cells covering the spatially barcoded spots were further categorized into four tissue types: (i) stroma, (ii) 1–10% epithelium, (iii)

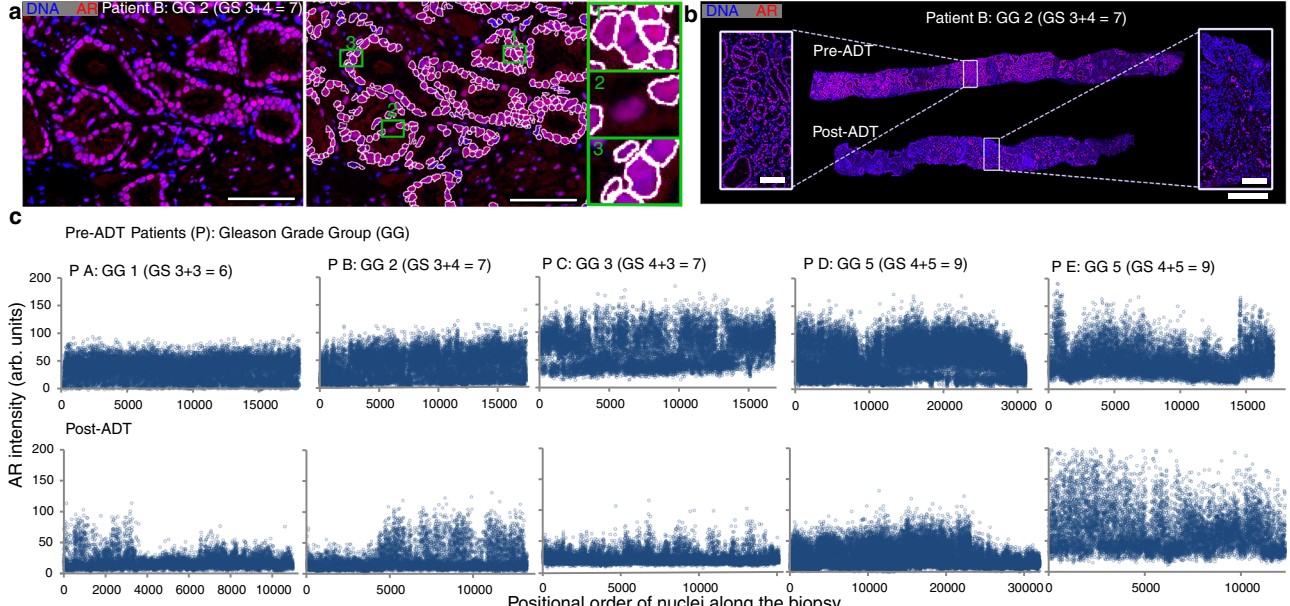

**Fig. 1 | IHC images for the AR activity in epithelial nuclei performed on needle biopsies pre- and post-ADT. a** Prostate tissue from a paraffin-embedded biopsy stained for the AR (red) and DNA (blue). AR levels in epithelial cell nuclei pre- and post-ADT were extracted, encircled in white (right panel; for details see Online methods). Scale bars 100 μm. Insets 1–3 show examples of occasional mis-classifications that may occur using the algorithm. Inset 1 shows failure of separating two nuclei. Inset 2 shows failure of nucleus detection due to low DNA-staining intensity. Inset 3 shows misclassification of stromal cell nuclei as epithelial, due to their proximity to epithelial cells. **b** Biopsies pre- and post-ADT from patient B. Small clusters of cells with nuclear AR still exist after ADT. Scale bar in whole figure, 1 mm, scale bars in close up, 100 μm. **c** Nuclear AR intensity from five patients with Gleason scores ranging from 6 to 9 (GG 1–5). The x-axis indicates the positional rank of each cell in the biopsy's length direction while the y-axis shows the mean intensity of the androgen receptor in each cell's nucleus. Source data is provided as Source Data file. AR androgen receptor, ADT androgen deprivation therapy, P X patient X, GG grade group, GS Gleason score.

11–50% epithelium, (iv) 51–100% epithelium. An overview of the collected data, annotations, and performed analyses is illustrated in Supplementary Fig. 4.

Each patient was assigned to a representative clinical response group regarding ADT-treatment: *responder* (patient 1), *moderate responder* (patient 2), and *non-responder* (patient 3), based on the clinical data (Table 1, Supplementary Table 2), such as PSA nadir, PSA progress, and metastasis status.

### Gene expression analysis in situ before and after ADT treatment

Gene expression analysis in tissue sections was achieved using barcoded arrays with spots on the surface, each with a known x- and y-coordinate. Each spot had a diameter of 100 micrometers, capturing the transcriptome from around 10–50 cells. The sample handling protocol for prostate tissue[55] was adjusted to be compatible with the limited amount of tissue provided by core needle biopsies (Supplementary Figs. 5–6). On average, we detected approximately 4000 expressed genes per spot for all biopsies (Supplementary Fig. 6c).

First, to obtain an overview of patients and their biopsies, we bulked all spots per biopsy into individual *pseudo*-bulk samples and performed a principal component analysis (PCA). Most of the biopsies separated patient-wise (Fig. 2b) as has previously been observed in patients with PCa[56]. For patient 1 (responder), biopsies separated pre- and post-ADT. Patient 3 (non-responder) exhibited only a small separation, while patient 2 (moderate responder) showed more extensive spread between the biopsies.

To further assess the overall gene expression differences, we next conducted differential gene expression (DGE) analysis between histological areas comparing pre- and post-ADT samples. By merging gene expression data from epithelial and stromal spots, respectively, followed by DGE analysis, we could compare the temporal changes between the two histological entities. AR-regulated genes, such as *KLK3*, *KLK2*, and *NKX3-1*[57], were downregulated post-ADT in epithelial spots, for the three patients (Fig. 2c, Supplementary Fig. 7a–d). In line

with clinical responsiveness, patient 1 had more differentially expressed AR-regulated genes compared to the other patients, indicative of successful and persistent downregulation of AR by ADT.

To identify biological pathways associated with differentially expressed genes (DEGs, $q < 0.01$), we performed functional enrichment analysis, querying against the KEGG database[58] (Fig. 2d, Supplementary Fig. 7e–g). Among the three patients, pre-ADT, pathways related to AR (e.g., PPAR signaling, steroid hormone biosynthesis, sphingolipid signaling) and protein processing (e.g., protein processing in endoplasmic reticulum, lysosome, phagosome) were activated. Pathways related to cell migration (regulation of actin cytoskeleton, focal adhesion) were activated post-ADT for all patients. Subsequently, the same analysis procedure was performed for stroma demonstration and an upregulation of immune response genes post-ADT in all patients was observed (Supplementary Fig. 8).

### Spatially resolved transcriptomes of patient biopsies

Next, we investigated the spatially resolved transcriptomes for the study cases. In total, we generated spatial and transcriptome-wide data for more than 4000 barcoded spots from 48 core needle biopsy sections including two consecutive tissue sections per biopsy. ST data from patients were analyzed individually by applying Spatial Transcriptome Decomposition (STD)[52] (Supplementary data files 1–6; schematically overviewed in Supplementary Fig. 2). STD is a probabilistic model that factorizes the observed transcript data into latent gene expression factors (Methods). The factors characterize distinct metagenes, groups of genes that are likely to be co-expressed, and their spatial expression patterns.

The patient-specific factor analysis provided 13, 16, and 10 gene expression factors for patient 1, 2, and 3, respectively (Methods), and a control was made to ensure the independence between the factors (Supplementary Fig. 9). The ranked list of marker genes within each factor was used for molecular annotation. We broadly categorized factors into three entities; stroma, immune enriched stroma, and

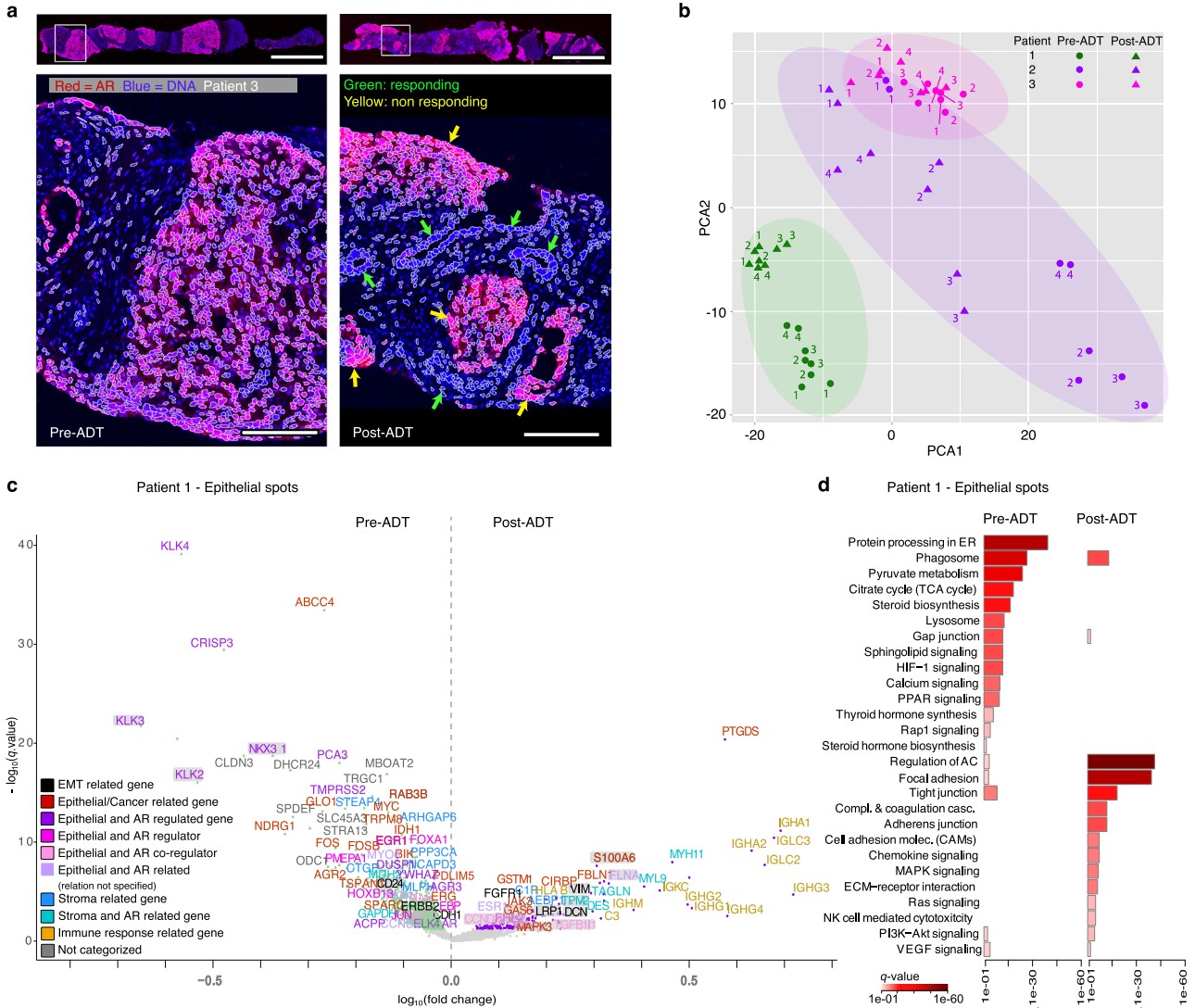

**Fig. 2 | Overview of AR activity and bulk analysis results pre- versus post-ADT.**
**a** Nuclear AR activity pre- and post-ADT identifies responding and resistant prostate cancer cells post-ADT (green and yellow arrows, respectively). The scale bars in the upper panels are 1 mm, in lower panels 200 μm. **b** Principal component analysis of *pseudo*-bulked spots per needle biopsy pre- and post-ADT, after rlog transformation. Source data is provided as Source Data file. **c** DGE analysis (Welch's *t*-test) of epithelial spots post- versus pre-ADT in patient 1. AR regulated genes are significantly down-regulated post-ADT (magenta). Genes shaded gray are relevant to PCa or genes of interest. Source data is provided as Source Data file. **d** Activated pathways for genes upregulated post- and pre-ADT (*q* < 0.05), color scale represents significance. ADT androgen deprivation therapy, AR androgen receptor, DGE differential gene expression.

tumor activity (Supplementary Figs. 10–12). Factors representing normal epithelium were not identified. Overall, the molecular annotation of the factors overlapped with the histological annotations, but the factor analysis provided an increased resolution in the analysis. For example, multiple tumor factors were identified enabling the temporal subdivision into responding or non-responding factors. These factors were then compared with AR staining in adjacent sections.

Hierarchical clustering of the factors was performed for the patients (Supplementary Figs. 13–16) providing a summary of the factors, the corresponding histological entities and the molecular annotations for the tumor factors. The overview also depicts the annotation of temporal differences in AR staining.

The UMAP visualization of factor activities and its spatial position in the core needle biopsies is shown in Fig. 3. The two-dimensional embedding (UMAP) presents the distinct factors that correspond to histological features such as stroma, and tumor. In addition to the temporal differences observed for the tumor factors, we identified stroma factors that also displayed temporal differences.

The molecular heterogeneity as determined by the number of tumor factors and spread in UMAP space (left panels) appeared to be most prominent for patient 3. In patient 3, at least eight distinct tumor factors could be identified while patient 1 displayed five tumor factors. The UMAP results for patient 2 points to a broader representation, represented by eleven tumor factors, in line with the initial PCA data on pseudo-bulk data on tissue sections, which indicate a higher degree of tissue heterogeneity.

The spatial position of the tumor factors along the needle biopsies outlines the temporal differences after eight weeks of ADT, raising certain observations from our analysis; Patient 1, the treatment responder, has a reduction of tumor factors post-ADT, where they largely are replaced by stroma factors (right panel). Interestingly, even though this patient responded clinically, as determined by a persistent low PSA-value, we can still observe some tumor factors post-ADT. We cannot determine whether the post-ADT tumor cells that express these factors are in a transient mode of disappearing, still capable of having AR-activity, or represent pre-existing resistant cells.

**Table 1 | Clinical data for the three patients used in the study**

|  | Patient 1 responder | Patient 2 moderate responder | Patient 3 non-responder |
|---|---|---|---|
| Initial status |  |  |  |
| Clinical stage | Local advanced, non-metastatic PCa | Local advanced, initial bone metastases | Local advanced, bone metastases and progression to retroperitoneal lymph nodes |
| PSA [ng/mL] | 41 | 10780 | 65 |
| Skeletal scintigraphy | Negative | Positive | Positive |
| After a minimum of 8 weeks of GnRH-analogue |  |  |  |
| PSA nadir [ng/mL] | 0.17 | 2.1 | 17 |
| CRPC[a] | No | Yes | Yes |
| Clinical cancer relapse |  |  |  |
| PSA relapse [ng/mL] | Negative (<0.1) | Positive (90) | Positive (1033) |
| 5-year survival | Yes | Yes | No |

*PSA* prostate specific antigen, *GNRH* gonadotropin releasing hormone, *PCa* prostate cancer, *CRPC* castration-resistant prostate cancer.
[a]CRPC defined as two consecutive rises in PSA while on ADT with castration-levels of testosterone <50 ng/dL)[147].

In patient 2, the moderate responder with high initial and low PSA levels at week eight, we observe four responding and seven non-responding factors. However, we observe a major change in tissue makeup post-ADT, similar to Patient 1. Most of the tissue after treatment is composed of stroma and immune response genes.

Patient 3, the non-responder with moderate PSA levels before and after eight weeks, had no responding factor but eight non-responding factors. The spatial distribution and amount of the tumor factors were not changed for Patient 3, over the therapy period. Although patient 3 only displayed non-responding factors, we detected a down-regulation of AR-regulated genes when comparing epithelial spots pre- and post-ADT (Supplementary Fig. 7c). This suggests that spots designated to non-responding factors also contain a fraction of cells that do respond to ADT.

Overall, the ratio of responding and non-responding tumor factors after eight weeks of treatment agrees with the clinical outcome of ADT (Supplementary Fig. 13). In all three patients, the tissue area corresponding to identified tumor factors post-ADT, and thus, the number of cancer cells, decreased substantially due to the apoptotic effect of ADT[59]. Responding factors were overall present in larger groups of spatially confined spots pre-ADT than non-responding factors (pre-ADT) (Supplementary Figs. 10–12). This is also true for the non-responding tumor factors, which are generally present in larger areas pre-ADT compared to a more dispersed pattern post-ADT, except for patient 3 (Fig. 3, Supplementary Fig. 17).

Further, we observed in the PCA analysis (Fig. 2b) that biopsy 1 (pre-ADT) for patient 2, overlaps with non-responding factors in patient 3, which might indicate presence of a resistant phenotype in patient 2.

Neuroendocrine differentiation (NED) can occur in prostate cancer. Prostatic adenocarcinomas that have undergone NED are resistant to ADT. To investigate if there was an enrichment of cells that had undergone NED in the areas expressing resistant factors, we stained all biopsies in patient 2, before and after ADT, with the neuroendocrine marker chromogranin A (CgA). Areas expressing a given factor were mapped against the corresponding area in the CgA stained section, and the ratio of CgA positive cells for the areas was quantified. The criteria for the area depicted was that at least five spots with a given factor should be clustered together in both ST-replicates. No difference in ratio of CgA positive cells was found in areas expressing resistant factors compared to areas expressing non-resistant factors. (Supplementary Figs. 18–19, Supplementary Tables 3–4).

To interpret the spatial RNA data obtained from the ST method it is of importance to know how well it correlates with the corresponding protein levels in the tissue. In a large study, including expression data from 60 genes in several different tissues and cell-lines, the correlation between the number of RNA transcripts and number of protein molecules was good within each gene, but less accurate when comparing the number of transcripts with the number of protein molecules in-between different genes[60]. In accordance with this result, we showed in a previous article, that there is a good concordance between RNA expression detected with ST technology and protein levels detected with immunocytochemistry for all 7 proteins tested in prostate tissue applying similar ST protocol used herein[55]. Here we show that this relationship between RNA expression and protein levels also is valid for the androgen receptor (Supplementary Fig. 20).

### Processes modified in non-responsive tumor factors

The factor annotation across the biopsy sections served as a tool to improve our investigation of the processes that underlie non-responsiveness. We, therefore, undertook a DGE analysis by first bulking the gene counts of the responding and non-responding factors, respectively, for patients 1 and 2. Here, spot selections were based on a stringent scheme: for example, spots with tumor factors above a defined factor intensity threshold were counted as non-responding spots if they consisted of histologically annotated tumor cells (see Methods for details, Supplementary Fig. 21a, b). Importantly, we only used information collected from spots pre-ADT to exclude ADT as a confounder in the DGE analysis.

The selection process resulted in an approximately equal number of responding and non-responding spots for patient 1 (71 and 90 spots, respectively, i.e., 56% non-responding spots from a total of 161 spots), while for patient 2 the fraction of non-responding spots was higher (64% non-responding spots from a total of 254 spots). The top differentially expressed genes (DEGs, $q$-value < 0.05 and $log$FC > 0.3 for patient 1 and > 0.5 for patient 2) between responder and non-responder spots were visualized as a gene expression heatmap (Fig. 4a, b). Only four DEGs were identified in patient 1, while patient 2 had more than 50 DEGs, potentially reflecting the clinical responsiveness of the patients.

To check that the DE-results of patient 1, which are immune related genes, are belonging to the non-responding factors and not simply is the result of non-factor-related cell types (since we are doing the analysis on all transcripts in the selected spots), we plotted all factors in patient 1 for their $log_2$ fold changes for these specific genes (Supplementary Fig. 22). The result shows that the non-responsive factor 11 has a FC between 2 and 3 for all these four genes.

For patient 2, spots from factor 5, 9, 10 and 14 were excluded from the DGE analysis, because of not fulfilling the specified criteria, such as

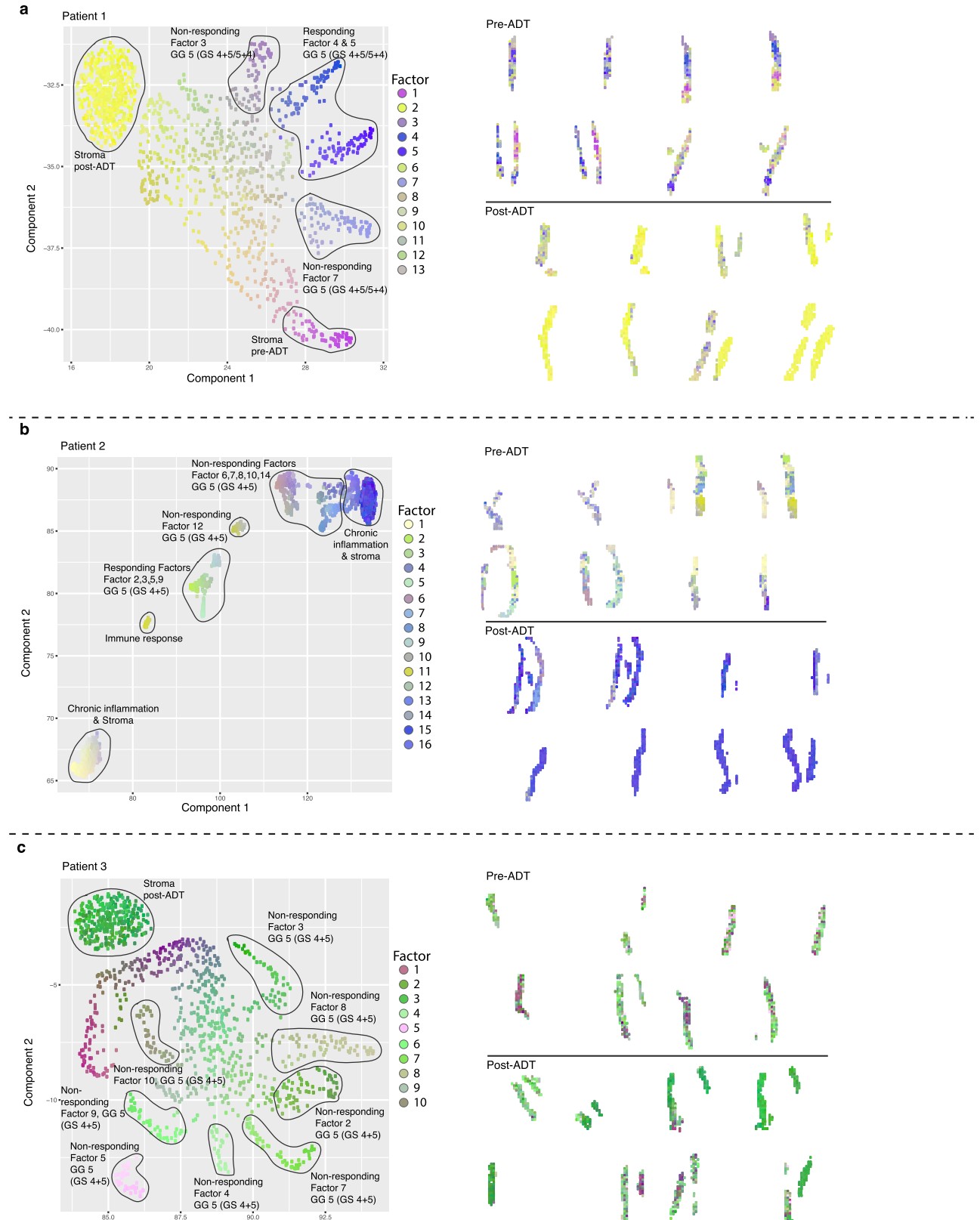

**Fig. 3 | UMAP visualization of gene expression factors (left) and the spatial position of gene expression factors (right). a** Patient 1- clinical responder. **b** Patient 2 - moderate responder. **c** Patient 3 - non-responder. Clusters of spots from factors of the same type are encircled. UMAP Uniform Manifold Approximation and Projection.

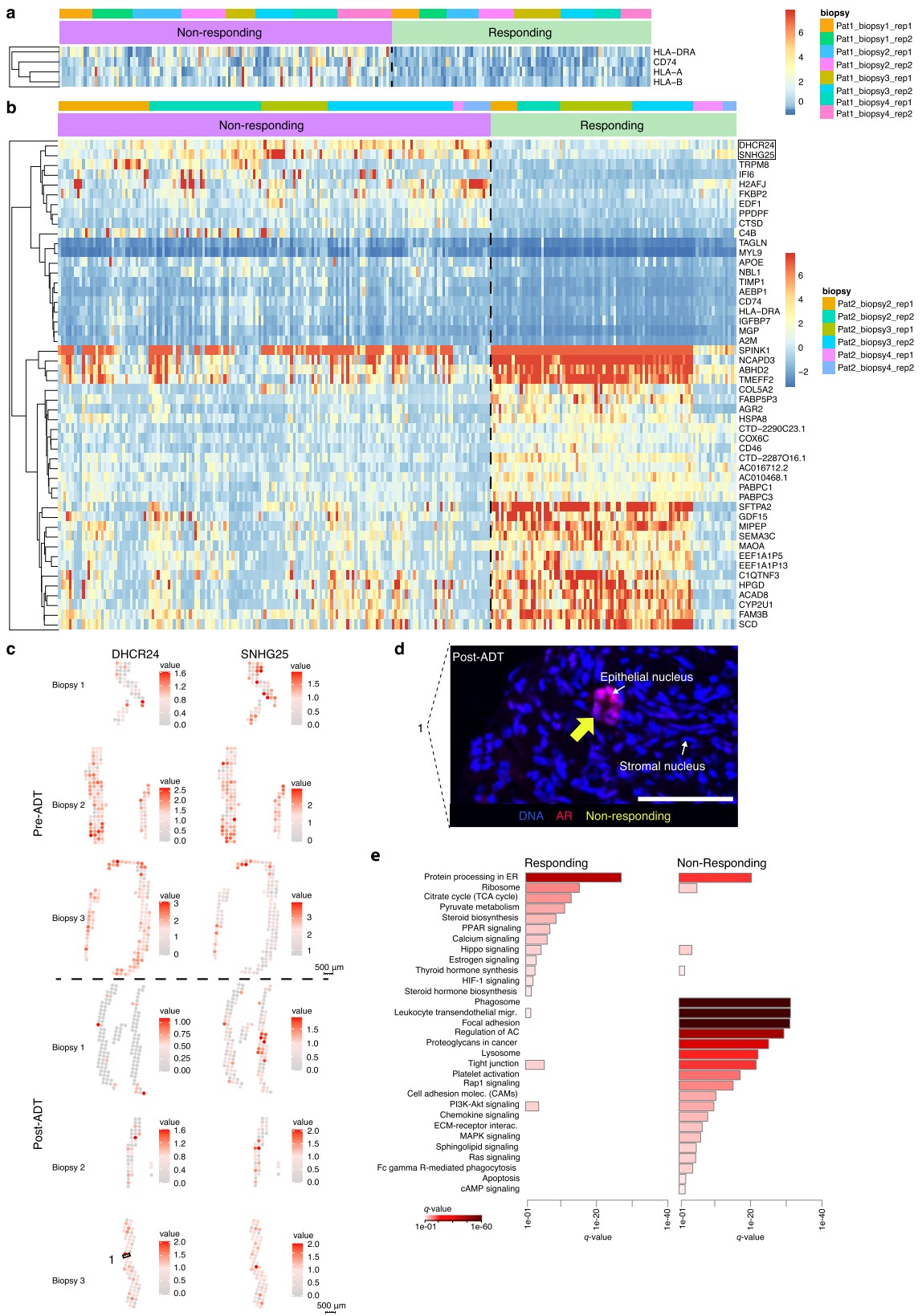

presence of factor activity in more than one biopsy (see Methods for details). The DEGs in patient 2 could be separated into two groups - one group of genes that are predominantly expressed in the non-responding spots and another group of genes with less distinct separation between responding and non-responding spots. The non-responding group of genes (including *DHCR24, SNHG25, TRPM8, IFI6,*

*FKBP2, PPDPF, HLA-DRA, TAGLN, TIMP1, AEBP1, IGFBP7, MGP, A2M,* and *CD74*) has previously been described to correlate with e.g., androgen-independence, cell migration, resistance, PCa, cancer, tumor stroma, immune response, and inflammation[61–74] (see Supplementary Tables 5–6 for complete list including downregulated genes). To spatially analyze the observations, the gene expressions of *DHCR24* and

**Fig. 4 | DGE analysis of responding versus non-responding prostate tumor factors. a** Tissue regions in Patient 1 (*q*-value < 0.05, *log*FC > 0.3), pre-ADT and **b** tissue regions in Patient 2 (*q*-value < 0.05, *logFC* > 0.5). The top 50 DEGs are shown and an overlap with patient 1 of non-responding genes *CD74* and *HLA-DRA* was observed. Scale-bars indicate normalized counts. Tissue regions were pre-ADT and a two-sided Wilcoxon rank-sum test adjusted for multiple comparisons was used. Source data are provided as Source Data files for **a** and **b**. **c** Gene expression of *DHCR24* and *SNHG25*, upregulated in non-responding cells, plotted onto tissue sections pre- and post-ADT in patient 2. **d** Representative example of nuclear AR activity in an epithelial luminal gland (yellow arrow), post-ADT, at an area where the non-responding gene *DHCR24* is highly expressed (marked with '1' in the bottom row in **c**) in patient 2. This serves as a confirmation of the presence of non-responsive cells in these areas. Scale bar 50 μm. **e** Activated pathways for DEGs upregulated in non-responsive spots or upregulated in responsive spots (*q* < 0.05) in patient 2, color scale represents significance. Source data is provided as Source Data file. ADT androgen deprivation therapy, AR androgen receptor, pat patient, rep replicate, DEG differentially expressed genes, FC fold change.

*SNHG25* were plotted onto the biopsy sections (Fig. 4c). Both genes are present over investigated time but more spatially prevalent before treatment onset.

To evaluate the set of predominantly non-responding genes, we took advantage of patient 3, the non-responder. Indeed, we confirmed that many of the marker genes (*DHCR24, TRPM8, IFI6, H2AFJ*) in non-responding areas in patient 2 were also expressed in the non-responder spots of patient 3 (Supplementary Fig. 23a). To ascertain that predicted areas with non-responding gene profiles are indeed non-responding we took advantage of the AR staining in the adjacent tissue sections, demonstrating nuclear staining for tumor cells in all areas post-ADT that had been annotated as non-responding. Some of these areas were crowded with eAR(+) nuclei and some had only sparse nuclei of this type. Non-responsive prostate glands and scattered cells, with nuclear AR, were found in areas where *DHCR24* and *SNHG25* were expressed, and a representative image of corresponding AR-stained tissue region shows a small AR-active prostate gland (Fig. 4d). Representative areas of AR-staining across the tissue sections post-ADT are shown in Supplementary Fig. 23b.

We continued by performing pathway analysis on all significant DEGs found in non-responding versus responding spots in patient 2 (Fig. 4e). Analysis revealed pathways associated with migration[75–77], survival[75], metastasis[75,76,78,79] and androgen-independence[75], proliferation[80], and angiogenesis[80] (focal adhesion, regulation of actin cytoskeleton, cell adhesion molecules (CAMs), proteoglycans in cancer, and platelet activation). Furthermore, pathways such as ECM-receptor interaction, MAPK signaling, RAS signaling, Rap1 signaling, PI3K-Akt signaling, and immune-related leukocyte transendothelial migration were more prominent in the non-responding spots.

The underlying non-responding genes give further insight into molecular processes. For example it has been demonstrated that primary tumor cells can stimulate platelets to get activated[81] which leads to the release of a wide range of growth factors and cytokines, such as TGFβ1[82] which has been shown to induce metastasis[83–86].

Platelet activation can also activate intracellular signaling cascades, such as p42 MAPK, which can stimulate proliferation, survival, adhesion and chemotaxis of hematopoietic cells[81]. Further, MAPK- and the PI3K-Akt pathways play a key role in apoptosis and bone metastasis[87]. MAPK-mediated phosphorylation of the nuclear receptor co-activator 1 (NCoA1, or SRC-1) may increase the coactivators affinity for AR, contributing to disease recurrence and CRPC[88].

The identified RAS/MAPK-pathway in the non-responding regions has been shown to contribute to PCa progression and metastasis[89], and is activated in 43% of primary tumors while in 90% of metastatic tissues[90]. RAS signaling has, in cell lines, shown to decrease androgen dependence and promote metastasis[91]. Several studies suggest that the PI3K-Akt pathway is involved in androgen-independent growth of PCa[92–97], and genetic alterations of PI3K-Akt pathway occur in 100% of metastatic PCa which suggest a key role in the progression to CRPC[12]. The PI3K-Akt signaling pathway is also known to induce stem-like properties, proliferation, migration, angiogenesis, regulation of cellular growth and survival[98], as well as having a potential correlation with PTEN-loss[99].

The enrichment of ECM-receptor interaction in non-responding areas has previously been shown to play a key role in metastasis since it needs to allow for a CAM-mediated coordinated balance between adhesion and detachments of tumor cells[100–102]. Akt signaling is a master regulator when it comes to inducing EMT and cancer stem cell phenotype by the ECM, and this can be mediated by various focal adhesion proteins and lead to activation of e.g., NF-κB[103,104]. Focal adhesion formations transduce ECM signaling into the tumor cells and activate the PI3K-Akt pathway[105].

Furthermore, the pinpointing of Rap1 signaling in non-responding areas is interesting as this has shown to induce cancer cell proliferation and disease progression in several cancer types[106–108], and particularly in PCa its activation affects integrins important in migration, invasion, and bone metastasis[109,110]. Increased Rap1 activity correlated with high metastatic potential in both PCa cell lines and in vivo, implicating Rap1 could be of therapeutic importance for curing PCa[110]. The leukocyte transendothelial migration is an important step in the initiation of an inflammatory immune response and chronic inflammation, suggested to serve as an anti-cancer therapy[111]. Further, for invasive cervical cancer, KEGG pathway enrichment analysis has revealed activated pathways such as 'focal adhesion', 'ECM-receptor interaction' and 'platelet activation'[112].

We also observe that these resistant tumor areas have a lower cell cycle activity as compared to non-resistant areas before treatment onset, when comparing to a 71 cell cycle gene signature[113] (Methods, Supplementary Fig. 24).

## The tumor microenvironment in prostate cancer

Previous studies have shown that lack or low levels of nuclear AR in stromal cells (sAR(-)), adjacent to tumor cells, are observed in the context of high Gleason scores, and metastasis[114,115]. In contrast, normal stroma expresses nuclear AR. Across our tissue sections, we observed multiple regions of sAR(-) cells (Fig. 5a) that was compared to our factor analysis. Hereby, we could investigate the stroma of responding and non-responding tumor factors, respectively.

The proportions of stromal AR-staining in stromal nuclei is illustrated as a pie chart per factor for each of the patient in Supplementary Figs. 14–16. We hypothesized that a responding tumor factor would have a higher extent of surrounding stromal AR-positive (sAR(+)) cells, while non-responding tumor factors would be encircled by sAR(−) cells.

All stromal tissue spots were annotated by their AR-content using a defined threshold where tissue signals below the cutoff were treated as AR(−) stroma. In short, spots, by visual detection, composed solely of nuclei that lacked AR were counted as sAR(−), and spots with nuclei with at least 50% nuclear AR, were counted as sAR(+) (described in detail in Methods). Areas with a mix of sAR(−) and sAR(+) spots were annotated to sAR(mix) (Supplementary Fig. 21c).

For patient 1, 72% of the 287 epithelial-containing spots with attributed non-responding tumor factors were associated with sAR(−), while 15% were associated with sAR(+) (the remaining spots contained a mixture of AR positive and AR negative cells). 27% of the 79 epithelial-containing spots with attributed responding tumor factors were associated with sAR(−), while 60% were associated with sAR(+) (the remaining spots contained a mixture).

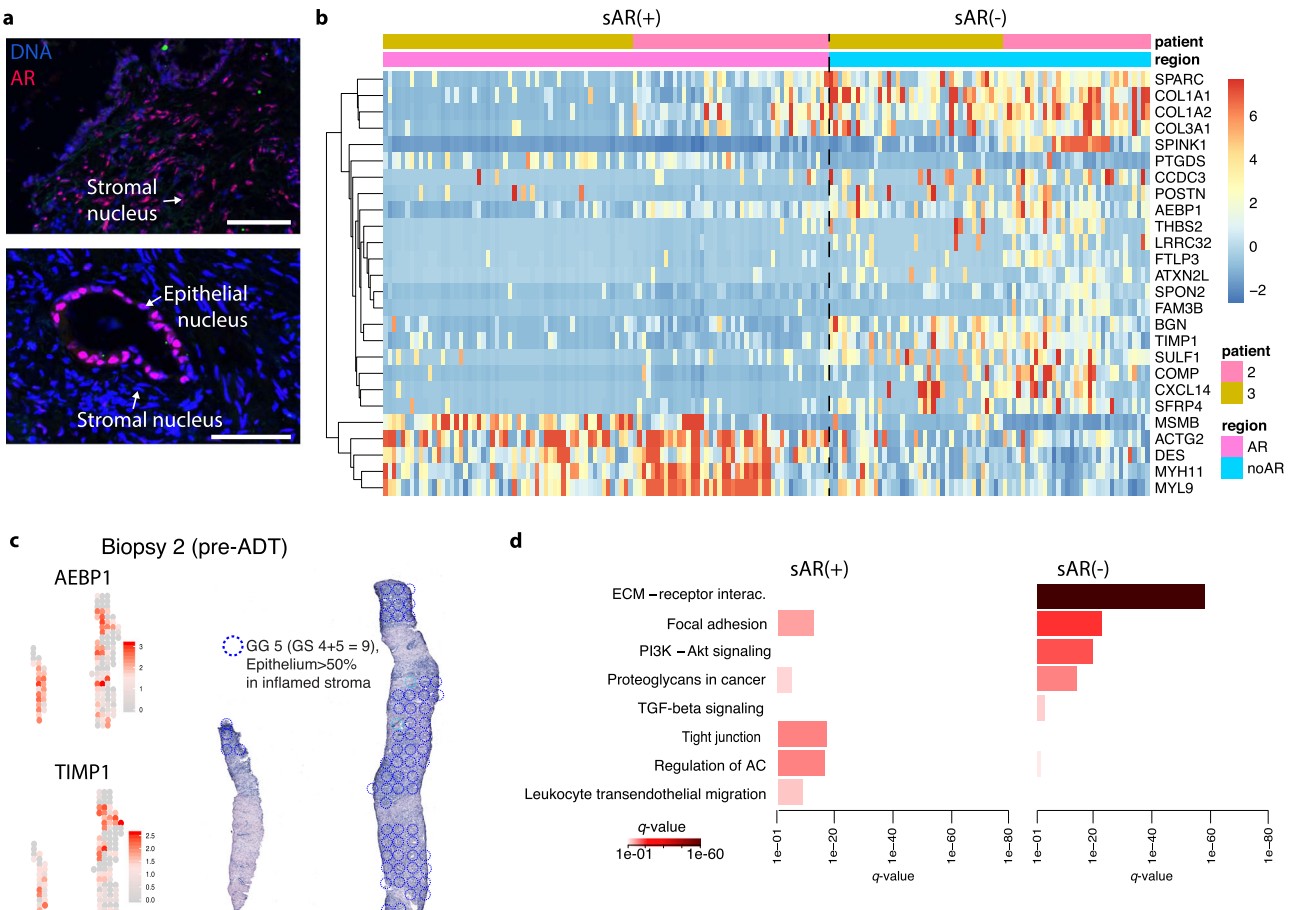

**Fig. 5 | AR(+) stroma versus AR(−) stroma in biopsies pre-ADT. a** Representative images of sAR(+) (top) and sAR(-) (bottom), in patient 2. Note the AR(+) luminal epithelial cells of the gland (bottom). Scale bars 50 μm. **b** Heatmap of the 26 DEGs (two-sided Wilcoxon rank-sum test adjusted for multiple comparisons, *q*-value < 0.05, *log*FC > 0.5) in the stroma with and without nuclear AR across spots in patient 2 and 3, respectively. Source data is provided as Source Data file. **c** *AEBP1* and *TIMP1* expression in a tissue section from patient 2 (pre-ADT, biopsy 2, left). HE-image with encircled spots annotated to GG5 and cell type composition of epithelial cells >50% (right). **d** Pathways activated for DEGs upregulated in sAR(−) areas or upregulated in sAR(+) areas (*q* < 0.05), color scale represents significance. Source data is provided as Source Data file. GG Grade group, GS Gleason score, ADT androgen deprivation therapy, GG Gleason grade group, GS Gleason score, AR androgen receptor, sAR(+) stromal nuclear AR positive, sAR(−) stromal nuclear AR negative, DEG differentially expressed genes, FC fold change.

For patient 2, >99% of the 424 epithelial-containing spots with attributed non-responding tumor factors were associated with sAR(−). 53% of the 61 epithelial spots with attributed responding tumor factors were associated with sAR(−), and 47% contained a mixture.

For patient 3, 81% of the 399 epithelial-containing spots with attributed non-responding tumor factors were associated with sAR(−), and 9% were associated with sAR(+). No responding factors were found in patient 3.

We continued to investigate sAR(−) areas in patient 2 and 3 in more detail. We sought to compare gene expression levels between the stromal tissue regions sAR(+) and sAR(−) located next to tumor areas, independent of responsiveness of potential nearby tumor factors. DGE analysis was performed on the expression data between spots belonging to each tissue type (Fig. 5b, Supplementary Fig. 21d). The fraction of sAR(−) spots was 42% of a total of 172 spots. 26 DEGs were identified of which 21 was upregulated in sAR(-) spots (Supplementary Table 7), for example the epithelial- mesenchymal transition (EMT)-associated genes *COL1A1, COL1A2, COL3A1, BGN, POSTN, SPARC,* and *AEBP1*[116–118]. *BGN* is also associated with poor prognosis and *PTEN* deletion[119] and upregulation during tumor angiogenesis[120], and *SPARC* promotes bone metastasis in prostate cancer[121]. Both *SPARC* and *POSTN* are glycoproteins important for the structural network in the ECM. *POSTN* has been shown, using in vitro models, to be upregulated in advanced stages of cancer stroma and in bone metastases, however

not in advanced stages of tumor cells[122], in line with our observations (Fig. 5b, Fig. 4b). Further, *SFRP4* is a marker for aggressive PCa and also a post-surgery recurrent marker[123], and *TIMP1* expression has been shown to be elevated in PCa stroma, to stimulate cancer associated fibroblasts, and to promote tumor progression[124]. The genes *AEBP1* and *TIMP1* are plotted onto the tissue section of biopsy 2 from patient 2 (pre-ADT) (Fig. 5c), and a comparison with histology reveals that these stromal compartments are located adjacent to PCa cells annotated as GG5.

Finally, in sAR(-) regions, we found an upregulation of several interesting pathways (Fig. 5d), of which the four most prominent ones were also upregulated in the non-responding tumor factors (ECM-receptor interaction, focal adhesion, PI3K-AKT, and Proteoglycans in cancer; Fig. 4b), together with TGF-β. The ECM-receptor interaction pathway has previously been shown to correlate with high reactive stroma content in PCa[125]. Also, changes in proteoglycans in the tumor microenvironment occur during tumor progression and affect e.g. cell signaling, chemokines, growth factors, and apoptosis[126]. The fact that the TGF-β pathway was upregulated in the sAR(-) regions is of particular interest since this can be induced by platelet activation which was activated in the non-responding tumor areas. TGF-β signaling promotes tumor initiation, progression, metastasis, EMT, stroma-tumor crosstalk, inflammation, immune-response, and angiogenesis[81–86,127,128]. Also, upon dissemination to the bones, tumor cells activate osteoclasts

to degrade the bone matrix and release the stored TGF-β, which in turn leads to enhanced tumor cell malignancy[129].

In summary, albeit sparseness of the material in the needle biopsies, we observed that a majority of the stroma in proximity to non-responding cancer cells pre-ADT lacks AR to a higher extent, in all patients. We noted that patient 1, who clinically had the best response to the treatment, displayed more sAR(+) surrounding the cancer, independent of responsiveness, while patient 2 and 3, who developed CRPC, displayed a higher frequency of sAR(−) areas in proximity to the non-responding epithelial spots. Future validation of these findings is important to reveal biomarkers and drug targets connected to stromal changes during the development of CRPC.

## Discussion

Castration-resistant prostate cancer (CRPC) remains incurable with a need to identify actionable targets to enable the development of long-lasting treatment regimes. Androgens and androgen receptors are key drivers in the development and progression of this disease and have therefore been the main targets of therapeutic treatments for years. The major hurdle to this type of cancer therapy is the inevitable emergence of drug resistance. Multiple mechanisms and alternative pathways have been associated with androgen-independent growth observed in CRPC. Despite recent developments of treatments targeting AR signaling, CRPC remains terminal[130].

Until now, few studies have reported a comprehensive analysis of drug resistance associated with spatial and temporal heterogeneity. In this study, we identify from sequential biopsies, at near cellular level, several distinct subpopulations that either respond, or do not respond to ADT.

The intriguing question of whether ADT resistance is developed during the course of treatment or developed before ADT onset, has remained open. Here we identify tumor cells with castration-resistant potential present already before treatment and not as the result of evolutionary selection during ADT. Another study also shows that pre-existing castration-resistant PCa-cells exist in primary prostate cancer exists[131].

The rationale of the exclusion of evolutionary pressure is based on the time frame of eight weeks of ADT with a GnRH agonist, which much likely is too short for the presentation of evolutionary pressure, considering that it takes around 4 weeks until castration level of testosterone is reached with this type of ADT. Therefore, the actual androgen depletion time is around 4 weeks per patient, and not 8.

Computational factor analysis was used to identify these hidden cell populations across the spatial landscape of taken biopsies before and after treatment, and the corresponding gene profiles together with the activity maps of each expression profile were used to assign characteristics such as ADT resistance. We validated the existence of the identified resistant cell populations in the investigated material by orthogonal AR staining, demonstrating the androgen receptor activity of cancer cells after ADT.

A possible explanation for some patients becoming rapidly resistant to ADT, while enduring the quality-of-life changes associated with castrate testosterone levels, could be that when most of tumor cells are eliminated during the course of treatment, resistant cells, already present, are enabled to proliferate and spread at a pace faster than would otherwise be possible. The spatial distribution of the non-responding spots ranged from sparsely scattered to clusters. The former could indicate intra-prostate cell seeding, and the latter - the presence of a tumor node consisting of resistant cells that has outgrown the responding cells − if present post-ADT. Our DGE analysis of non-responding gene expression areas provides some initial molecular detail and shows an upregulation of genes involved in migration, resistance, immunosuppressive function, inflammation, cancer, and tumor stroma (e.g., *H2AFJ*, *FKBP2*, *MGP*, *A2M*, and *IGFBP7*).

Not all tumors metastasize and there is an urgent need to identify biomarkers associated with aggressive disease. The expression phenotype of the non-responding cells resembled that of cells that have undergone a partial epithelial-mesenchymal transition. Similar phenotypes have been reported before at the periphery of a cancer focus and the interaction between cancer cells and reactive stroma has been suggested to drive this transformation[55,132]. We, therefore, hypothesize that the non-responding cancer cells are situated in the rim of cancer foci emphasizing the importance of the cancer-stromal interaction for the evolution of resistant phenotypes (as outlined in Supplementary Fig. 25). We, therefore, suggest that further efforts should focus on investigating reverse epithelial-mesenchymal-transition and the interplay between reactive stroma and cancer foci. This could open up the possibility of a common treatment across CRPC affected men.

Furthermore, being biopsy-based, our findings could translate to clinical utility if drug resistance can be predicted by sampling the tumor before the commencement of treatment. The identification of castration-resistant cancer populations in hormone naïve PCa opens up the possibility of early selection of men for non-AR directed therapies such as chemotherapy or PARP inhibitors. Patient-derived explant methods or xenografts might offer an opportunity to find the optimal drug combinations or new small molecules[133].

We further explored the supporting role of the microenvironment. The cancer-microenvironment interaction provides vast opportunities for future treatment of prostate cancer. Nuclear AR-negative stroma has previously been correlated with increased invasiveness and biochemical relapse[114,115,134]. In this study, we identified nuclear AR-negative stromal cells to correlate tightly with unresponsiveness to ADT.

To reduce the risk of relapse of PCa, future potential probably lies in a combined treatment of the tumor and its microenvironment. Co-treatment of stroma and tumor is an important concept in cancer therapy. In the case for PCa treatment however, an intrinsic problem with ADT is that the tumor epithelial cells are the desired targets, but the stroma will also be targeted, which will induce a more lethal microenvironment since AR in stroma is required for a healthy phenotype. To overcome this, drugs could be developed targeting only epithelial cells by decreasing their AR signaling, which could be achieved by targeting AR co-regulators and pioneer factors that are specific for only epithelial cells, as suggested earlier[135].

In conclusion, this study presents a high spatial whole transcriptomic analysis of biopsies before and after ADT. We want to highlight that the power of using ST lies showing the spatial locations of cell clusters with similar gene profiles. The biological question would not be possible to answer with, e.g., single-cell RNA-seq alone, since the very nature of the technology eliminates the spatial integrity. Moreover, with ST, it is possible to identify the spatial distribution of the gene expressions across the tissue sections, meaning we can e.g. determine if a factor is sparse and spread out or located at one single cluster. Further, the neighborhood of tumor cells can be analyzed within the same experiment. Here we identified tumor cells with castration-resistant potential that are present already before treatment, and characterized potential biomarkers that may provide the advantage of being more specific and effective in future clinical management of PCa. We also characterized the gene expression of resistant cells' neighboring nuclear AR-negative stromal cells. Overall, we demonstrate the importance of a combined temporal and spatial analysis of tumor in the context of its microenvironment suggesting a new course of action to understand treatment resistance.

## Methods
### Ethics declaration
The study was performed according to the Declaration of Helsinki, Basel Declaration and Good Clinical Practice. The study was approved by the Regional Ethical Review Board (REPN) Uppsala, Sweden before

study initiation (Dnr 2011/066/2, Landstinget Västmanland, Sari Stenius). All human subjects were provided with full and adequate verbal and written information about the study before their participation. Written informed consent was obtained from all participating subjects before enrolment in the study.

## Study design
In brief, eight patients diagnosed with advanced prostate cancer were enrolled in this study, whereas three patients were analyzed with the ST technique. A total of eight prostate core needle biopsies were obtained per patient, in which four of the biopsies were taken before treatment initiation of androgen deprivation therapy (pre-ADT) and the remaining four biopsies eight weeks after (post-ADT).

## Array production
For tissue optimization arrays, poly-20TVN capture oligonucleotides (IDT) were uniformly printed on the surface of Codelink Activated microscope glass slides (#DN01-0025, Surmodics) according to the instructions of the manufacturer (Surmodics). The oligonucelotides immobilized on the surface were:

Reverse transcription oligonucleotide:

[AmC6] UUUUUGACTCGTAATACGACTCACTATAGGGACACGAC GCTCTTCCGATCTNNNNNNNNNTTTTTTTTTTTTTTTTTTTTVN

For spatially barcoded arrays, 1007 circular areas (spots) were printed onto the surface of Codelink Activated microscope glass slides (#DN01-0025, Surmodics) according to the instructions of the manufacturer (Surmodics). Each spot contained millions of uniformly spread oligonucleotides containing poly-20TVN capture regions (IDT) with a unique spatial barcode, serving as x- and y-coordinates incorporated into the DNA sequence, making it possible to trace back each transcript to its original histological position. Each oligonucelotide also had a 18-mer unique barcode, and a 9-mer semi-randomized UMI. The spot diameter was 100 μm and the center-to-center distance between two spots was 200 μm.

The spatially barcoded reverse transcription oligonucelotide immobilized on the surface was:

[AmC6]UUUUUGACTCGTAATACGACTCACTATAGGGA-CACGACGCTCTTCCGATCT[18MER_SPATIALBARCODE] WSNNWSNNVTTTTTTTTTTTTTTTTTTTTTTVN

Each array, both tissue optimization and barcoded ones, is 6200 × 6600 μm in size and the barcoded arrays contain a frame consisting of 148 spots with a specific oligonucleotide (Eurofins), enclosing the spatially barcoded spots, to retain the orientation.

## Tissue handling
Prostate core needle biopsies (12 × 0.6 mm) were placed on a piece of paper in a tube and stored at −80 °C at the site of surgery. Pre-ADT biopsies were taken systematically from the outer layer (peripheral zone) and rebiopsing were made from known tumor locations. The biopsies were then embedded individually or pairwise in cold OCT (#4532, Sakura). Cryosections were taken at 10 μm thickness and placed on spatially barcoded ST microscope glass slides as previously described[51]. Additionally, adjacent tissue sections were placed on standard microscope glass slides for immunohistochemistry analysis.

## RNA-seq library preparation and sequencing
We followed the protocol described previously[50,51], with exception to fixation, permeabilization and tissue removal conditions. Tissue sections were placed onto arrays covered with spot-patterned reverse transcription oligos, and then incubated at 37 °C for 1 min to attach the tissue. Then, a mild fixation was performed using 4% methanol-free formaldehyde (#28908, Thermo Fisher Scientific) diluted in 1xPBS (#09-9400, Medicago) for 10 min, followed by a wash with 1xPBS. The tissue sections were then stained with Mayer's Hematoxylin (#S3309, Dako) for 6 min, bluing buffer for 2 min (#CS702, Dako), followed by

Eosin (#HT110216, Sigma-Aldrich) in Tris-base (0.45 M Tris, 0.5 M Acetic acid, pH 6.0) for 30 s. Sections were then rinsed and dried, mounted with 85% Glycerol (#104094, Merck Millipore) and covered with a coverslip ("BB024060A1, Menzel-Gläser). Then, the arrays were brightfield imaged using Metafer Slide Scanning Platform (Metasystems). Images were stitched using the VSlide software (Metasystems).

Subsequently, the tissue sections were permeabilized by first incubation in 70 μl of 1x Exonuclease I Reaction buffer (#B0293S, NEB) with 0.19 μg/μl BSA (#B9000S, NEB) for 30 min at 37 °C, followed by incubation of 70 μl 0.013% pepsin (#P7000-25G, Sigma-Aldrich) dissolved in 0.1 M HCl (#318965-1000ML, Sigma-Aldrich) for 10 min at 37 °C, to enable diffusion and binding of the mRNA transcripts onto the barcoded array surface. In between and after these steps a wash with 100 μl of 0.1× SSC, diluted in RNase and DNase free water.

Next, reverse transcription was performed overnight, also in 70 μl, at 42 °C. The reverse transcription mix contained 1x First strand buffer (#18080-044, Invitrogen), 5 mM DTT (#18080-044, Invitrogen), 500 μM of each dNTP (#R0192, Fisher Scientific), 0.19 μg/μl BSA, 50 ng/μl Actinomycin D (#A1410-2MG, Sigma Aldrich), 1% DMSO (#472301-500 ML, Sigma-Aldrich), 20 U/μl Superscript III (#18080-04, Invitrogen) and 2U/μl RNaseOUT (#10777-019, Invitrogen).

Tissue removal was then performed in two steps to be fully efficient. First using 1% β-mercaptoethanol (#444203, CALBIOCHEM) dissolved in RLT buffer (#79216, Qiagen) for 1 h at 56 °C, and then by incubation of Proteinase K dissolved in PKD Buffer for 1 h 15 min at 56 °C.

The barcoded cDNA-products were then cleaved from the surface using a 70 μl cleavage mix containing 1.1x Second Strand Buffer (#10812-014, Invitrogen), 8.75 μM of each dNTP, 0.20 μg/μl BSA and 0.1U/μl USER enzyme (#M5505), NEB), that was incubated for 2 h at 37 °C. Until the cleavage of surface probes, the protocol were necessarily performed separately for each of the three patients, which potentially could introduce unwanted variation into the data.

Finally, 65 μl cleavage-mix was saved per sample for the upcoming library preparation that was conducted for all samples simultaneously. The samples were processed into sequencing libraries as described earlier[136] with the following steps: 5 μl of a second strand mix containing 2.7× First Strand Buffer, 3.7U/μl DNA polymerase I (#18010-017, Invitrogen) and 0.18U/μl RNaseH (#18021-014, Invitrogen) was added to the samples and incubated for 2 h at 16 °C. Then, 5 μl T4 DNA polymerase (#M0203S, NEB) was added and samples incubated 20 min. 25 μl of RNase and DNase free water was added followed by purification using Agencourt RNAClean XP beads (#A63987, Beckman Coulter) according to manufacturer's protocol and elution was made in RNase and DNase free water. Then, 5.6 μl of each sample was mixed with 10.4 μl In Vitro Transcription mix with a final concentration of 1x T7 Reaction Buffer (#AM1333, Ambion), 7.5 mM of each NTP (#AM1333, Ambion), 1× T7 Enzyme Mix (#AM1333, Ambion) and 1U/μl SUPERaseIN (#AM2694, Ambion) and incubated for 14 h at 37 °C. Sample purification was performed using Agencourt RNAClean XP beads according to manufacturer's protocol. Elution was made with 10 μl RNase and DNase free water.

Library quality control of the amplified RNA included measurement of average lengths of libraries using the RNA 6000 Pico kit (#5067-1513, Agilent) with a 2100 Bioanalyzer (Agilent) and concentration using Qubit dsDNA HS Assay Kit (Life Technologies) according to instructions of the manufacturer. The remaining sample and 2.5 μl Ligation adapter (IDT) was added to reach a final concentration of 0.71 μM. Then, the samples were heated for 2 min at 70 °C before placed on ice and then 4.5 μl ligation mix was added to a mix of 1x T4 RNA Ligase Reaction Buffer (#B0216L, NEB), 20U/μl T4 RNA Ligase2, truncated (#M0242L, NEB), 4U/μl RNase Inhibitor, Murine (#M0314L, NEB), and 0.5 μM Ligation adapter. The samples were incubated for 1 h at 25 °C. Then, purification of samples using Agencourt RNAClean XP beads as previously described was performed

followed by addition of 1 µl RT-primer (IDT) to reach a final concentration of 1.7 µM and 1 µl dNTPs to reach a final concentration of 0.83 mM of each dNTP. Samples were heated for 5 min at 65 °C, placed on ice, and then 8 µl reverse transcription mix was added to a final concentration of 1× First Strand Buffer, 0.05 M DTT, 500 µM of the each dNTP, 1 mM RT-primer, 10U/µl Superscript III and 2U/µl RNase-OUT. Samples were incubated for 1 h at 50 °C for 1 h, placed on ice, purified using Agencourt RNAClean XP beads as described above. Then, a total reaction volume of 10 µl with 1x KAPA HiFi HotStart ReadyMix (#KK2601, KAPA Biosystems), 1x EVA green (#31000, Biotium), 0.5 µM PCR lnPE1.0 (Eurofins), 0.01 µM PCR lnPE2.0 (Eurofins), 0.5 µM PCR Index (Eurofins) and 2 µl purified cDNA were amplified using qPCR with the following protocol: 98 °C for 3 min, then cycling at 98 °C for 20 s, 60 °C for 30 s and 72 °C for 30 s. Based on the qPCR-results, a suitable number of PCR-cycles was then used to amplify the samples using a total reaction volume of 25 µl. The samples were purified as described above and eluted in 20 µl elution buffer (#19086, Qiagen).

The libraries were quality controlled by measuring the average fragment length using the DNA 1000 kit (#5067-1504, Agilent) with a 2100 Bioanalyzer according to manufacturer's instructions. Library concentrations was determined using Qubit dsDNA HS Assay Kit (#Q32854, Life Technologies) according to the manufacturer's instructions. Finished libraries were diluted to 4 nM and sequenced on an Illumina Nextseq 500 instrument with v2 chemistry using paired-end reads, with a High Output-kit. On read 1, 31 bases were sequenced, which included the spatial barcode and a unique molecular identifier, and on read 2, 46 bases were sequenced to obtain genetic information of the captured transcripts.

To visualize the frame on the arrays, 70 µl hybridization mix with 0.96× PBS, 0.2 µM Cy3_anti_A_probe (Eurofins) and 0.2 µM Cy3_anti_Frame_probe (Eurofins) was incubated for 10 min at R, and subsequently washed with 100 µl 0.1× SSC, then 2x SSC with 0.1% SDS at 50 °C for 10 min, 0.2x SSC at RT for 1 min, 0.1x SSC at RT for 1 min and spin-dried. Then the arrays were mounted with SlowFade Gold Antifade Reagent (#S36963, Invitrogen) and topped with a coverslip. Imaging and stitching was perfomed as described above regarding bright field images.

Ligation adapter:
[rApp]AGATCGGAAGAGCACACGTCTGAACTCCAGTCAC[ddC]
Second reverse transcription primer:
GTGACTGGAGTTCAGACGTGTGCTCTTCCGA
PCR primer lnPE1.0:
AATGATACGGCGACCACCGAGATCTACACTCTTTCCCTA-CACGACGCTCTTCCGATCT
PCR primer lnPE2.0:
GTGACTGGAGTTCAGACGTGTGCTCTTCCGATCT
PCR index primer:
CAAGCAGAAGACGGCATACGAGATXXXXXXGTGACTGGAGTTC
Array hybridization oligonucleotide Cy3 anti-A probe
[Cy3]AGATCGGAAGAGCGTCGTGT
Array hybridization oligonucleotide Cy3 anti-frame probe
[Cy3]GGTACAGAAGCGCGATAGCAG

## Quality control array experiments

To optimize tissue permeabilization conditions, fluorescent cDNA footprints were created. The reverse transcription mixture was identically used for the RNA-seq libraries used for optimization and spatial capture, except for 0.5 mM of each dGTP/dATP/dTTP, 12.5 µM dCTP and 25 µM Cyanine 3-dCTP (#NEL576001EA, PerkinElmer) used in the Different combinations of pepsin concentration and pepsin times were tested after the Exonuclease I buffer-incubation. We tested a concentration of 0.013%, 0.025%, and 0.050% pepsin together with permeabilization times of 6 and 11 min. Each combination was tested on duplicates and the cDNA footprint was scanned in a NimbleGen

microarray scanner (Roche) with 5 µm resolution and gain at 10%. The signal intensities were measured using GenePix Pro 5.0 Microarray Acquisition & Analysis software. Intensities minus background on 3 epithelium and three stroma areas were determined on each duplicate using GenePix Pro (87/91 in Brightness/Contrast) yielding six values per treatment. Background on each well was determined by taking the average of three intensity values on the background area. A boxplot summarizing the intensities for 3 areas of the epithelium and stroma after subtraction of the background was performed on the duplicates.

## Trimming, alignment and annotation of sequences

FASTQ files were processed using the ST Pipeline[137] (version 0.8.3). Longer homopolymers than 15 bases were removed from read 2. Then, quality trimming was performed based on BWA followed by removal of all reads shorter than 28 bases. The remaining reads were then mapped using STAR[138] (2.5.0b) to the human assembly GRCh38 Ensembl[139] (release 86) using default settings. HTseq-count[140] (version 0.11.3) was used to quantify the mapped reads using annotation mode intersection-strict with human assembly GRCh38 Ensembl (release 86). The annotated reads were then demultiplexed on their indexes together with their corresponding read 1, containing the spatial barcode and UMI. Of these, the reads that did not have a valid spatial barcode were discarded. Of the reads that after this has the same gene and spatial barcode the unique molecular identifier (UMI) was used to discard duplicates. Genes believed to be of non-interest were removed before further analysis, including *MALAT1*, ribosomal, mitochondrial, ambiguous genes, and *RP11*-reads.

## Spatial transcriptome decomposition

STD[52] is an unsupervised method requiring no prior reference expression data for decomposing hidden cell type/state mixtures across the spots. The core model for STD uses ST count matrices for computation and visualization of the spatial gene expression data. STD allows unwanted batch effects to be accounted for, if sources of these are included as covariates in the analysis. The covariates included were pre- and post-ADT, which tissue sections that were consecutive sections from the same biopsy (i.e. 'duplicates'), and the slide and array number each tissue section was placed upon. Duplicates were placed on different slides and on each array a pre-ADT biopsy was placed along with a post-ADT biopsy, in order to reduce potential batch effects.

The factor analysis uses Bayesian shrinkage to avoid overfitting the expression factors. Notably, when extraneous factors are included, their inferred baseline expression levels will be very low. Thus, extraneous factors do not worsen model fit but may make results less interpretable by, for example, introducing noise in visualizations. To accommodate for this fact, we initially overspecified the number of expression factors and then reran the analysis with the number of factors appropriate for our data. This approach avoids underfitting while maximizing the expressiveness and interpretability of the final model. We decided to include factors which had >5000 transcripts contributing to each factor. 5000 transcripts are estimated to equal 50–500 cells, which is thus the minimal number of cells we determined to qualify for this analysis.

## Dimensionality reduction

To jointly visualize the factor activity maps, we compressed the information across the factor activities by reducing it into three dimensions, using uniform manifold approximation and projection (UMAP)[141], via umap-learn R package (version 0.1.5) using default settings except for '$n\_neighbors = 25$'. Input to the UMAP-analysis was factor-proportions, i.e. the expected number of reads for each factor in each spot. The resulting data was scaled into the unit cube and used as RGB color coordinates for each spot, resulting in a joint spatial activity map for all factors, where similar color represent similar gene expression profile.

## Annotation of transcriptomic factors

The factors resulting from the patient oriented performed STD were annotated as stromal and epithelial and the latter further as responding or non-responding. Factors can represent cell types or states of solely epithelial cells, solely stroma cells, or a mix thereof. Two separate, of each other independent, strategies for determining responsiveness were applied. First, for each patient, hierarchical clustering of factors using the expected number of reads per gene was performed. Distances between factors were calculated using the Bray–Curtis dissimilarity via vegan R package (version 2.5.6), the Ward D2 criterion[142] was used to identify pairs of clusters for merging and to build the trees via stats R package (version 3.6.3). Clusters of factors were given a main annotation as either stroma, inflammation, or cancer, and then if necessary, also some additional notes. The annotations of the trees were based on the overlap with the pathologists' annotations and the top genes from each factor.

Cancer clusters were further annotated as responding or non-responding mainly based upon the temporal location of each factor on the tissue sections, and also to the overlap with the pathologists' annotations.

Then, to construct a strict categorization of responsiveness of the tumor factors for upcoming analyses performed, the below criteria were made as a separate categorization of the factors. The rationale for the criteria was that, since we are only looking at small parts of the entire prostate of each patient, we cannot be sure that the biopsies are representative for the whole prostate, and if the patients do really have the, by us, assigned treatment response (on the molecular level). Therefore, to determine what factors were to be considered responding and non-responding, we decided to include several requirements to be met, to increase the chance of including only true responsive factors (or mainly responsive ones) as well as non-responding factors, and to minimize the risk of including tissue spots belonging to non-clear areas regarding responsiveness.

Criteria for determination of responsiveness vs non-responsiveness
- Cutoff of factor activity intensity at a threshold of 110, corresponding to a factor activity of 50%. This was used to exclude spots with relatively low activity of a given factor.
- A minimum of 10 spots per factor
- Requirement of pre-ADT presence of 30–100% spots annotated to cancer. These spots can be either annotated as >10–50% epithelial spots or as stroma with PCa infiltration.
- For the spot-based DGE analysis on factors, described in Fig. 4, spots belonging to both responding and non-responding factors were found in minority, and discarded. Further, if several neighboring spots belonging to a specific factor are annotated to cancer, also neighboring minority non-cancer annotated spots were selected as well if belonging to the same factor.

Specific criteria for determination of responding factors
- Number of spots of a responding factor was required to have a majority of spots located pre-ADT, and maximum a few spots located post-ADT. This cutoff was arbitrarily set to be <15%.
- Spots should cover two or more of the in total four biopsies pre-ADT, alternatively cover only one biopsy, if a similar pattern is seen across both biological duplicates, to increase the chance of having biological relevance. Note that the spots in the biological duplicates most often cover different tissue areas, although close to each other.

Specific criteria for determination of non-responding factors
- If factor activity was observed both pre- and post-ADT, such a factor qualified as a "non-responder" factor if constituent spots were annotated as cancer pre-ADT, but not necessarily also post-ADT (because of the intrinsic difficulty in capturing all sparse cells post-ADT surrounded by other cells when performing annotation).
- If factor activity was seen only post-ADT, >50% of spots in the factor needed to be annotated by the pathologist as residual cancer.
- Number of spots of the factor post-ADT should be >15%.
- A factor was included as a non-responding factor if the factor had activity in a minimum of 10 spots in at least one biopsy post-ADT.
- For spot-wise factor analysis described in Fig. 4, qualifying spots are required to, post-ADT, be annotated by pathologist as either one or a mix of the following:

'PCa'
'PCa infiltration in desmoplastic stroma'
'Suspicious for residual PCa (if other spots from the same factor belongs to spots located in cancer areas)'
'Residual PCa cannot be excluded (if other spots from the same factor belongs to spots located in cancer areas)'
'No evident PCa (if other spots from the same factor belongs to spots located in cancer areas)'

## Immunohistochemistry

**On paraffin embedded material.** All needle biopsies that were analyzed and not used for ST-analysis, were collected, paraffin embedded, cut and finally stained for the androgen receptor (AR) with primary AR antibody:(N-20, Cat. # cs 816, SCBT, 1:500), Ku70 antibody: (E-5, Cat. # sc-17789, SCBT, 1:500), phosphorylated DNA-PKcs:(S2056, Cat# ab18192, Abcam, 1:750). Secondary antibodies used were donkey anti-Rabbit IgG (H + L) Highly Cross-Adsorbed, Alexa Fluor™ 555:(Cat. # A-31572, 1:500, ThermoFisher scientific, Molecular Probes) and donkey anti-Mouse IgG (H + L) Highly Cross-Adsorbed, Alexa Fluor™ 488 (Cat. # A-21202, 1:500; ThermoFisher scientific, Molecular Probes). The DNA was co-stained with the fluorescent DNA intercalator TO-PRO 3 Iodide (1:1000, Molecular Probes). Tiled images of whole biopsies were collected with a Zeiss 780 confocal system using a Plan Apochromat 20X/ 0.7 NA objective.

In the paraffin embedded material, data driven identification of epithelial nuclei were used in combination with mean AR signals within nuclei in order to distinguish AR active epithelial cells from AR-negative epithelial cells in images of biopsies, taken before and after ADT. In short, the algorithm, written in Java using the ImageJ package (Rasband, W.S., ImageJ, U. S. National Institutes of Health, Bethesda, Maryland, USA, https://imagej.nih.gov/ij/, 1997–2018), uses the DNA signal as a mask for nuclei extraction. Spatially solitary nuclei are considered as nuclei of stromal cells and are excluded. The area of every extracted nuclei is mapped on the original image containing the AR signal. The mean intensity of the AR signal and the geometrical gravity center is then calculated for all extracted nuclei. Especially within high-grade cancer the epithelial cells will be very dense, and overlap between nuclei are common. A water-spread function has been used to separate these overlapped nuclei. The extracted fraction of epithelial nuclei will contain nuclei from luminal epithelial cells, basal cells, and a small fraction of stromal cells that have been located in close vicinity to epithelial cell clusters. The luminal epithelial cells normally have AR in the nucleus, but the basal cells do not. Thus, in an untreated needle biopsy of a prostate, nuclei with high to almost no AR signal is expected (see Fig. 1).

**On snap-frozen material.** Tissues from needle biopsy sections of 10 micrometer on superfrost slides, stored at −80 °C were thawed at RT to be fixated with 3% freshly made paraformaldehyde in TBS for 10 min in RT. Tissues were then permeabilized for 10 min in TBS + 0.1% Triton-X100, rinsed three times in TBS for 5 min/rinse and blocked with 2% bovine serum albumin in TBS for 2 h. Incubation with the primary AR antibody:(N-20, Cat. # cs 816, SCBT, 1:500), primary chromogranin A

antibody: (LK2H10, Cat. # MA5-13096, Invitrogen, 1:150) was done overnight at 4 °C. After that, the tissue was rinsed 3 × 5 min with TBS before incubation with the secondary antibodies donkey anti-rabbit IgG (H + L) Highly Cross-Adsorbed, Alexa Fluor™ 647: (Cat. # A-31573, 1:500), and donkey anti-mouse IgG (H + L) Highly Cross-Adsorbed, Alexa Fluor™ 555: (Cat. #A-31570, 1:500) for 1 h at RT in darkness. DNA was counterstained with DAPI (Molecular Probes) and slides were mounted with Prolong Gold (Molecular Probes). Fluorescence images were obtained using a Zeiss LSM 780 inverted confocal microscope, using a Plan Apochromat 20×/NA 0.7 objective. Tiled images were acquired from optical sections of 5 micrometers.

### Androgen receptor-based annotation of stroma

Immunohistochemistry (IHC) staining on the AR was performed on consecutive tissue sections to the ST-slides, followed by manual alignment enabling subsequent selection of gene expression data from spots of interest, under the anticipation that cells on both tissue sections had similar AR activity. In cases where alignment was not good directly, subareas of the tissue sections were aligned separately, in this way a manual correction for each part of the alignment of the tissue sections could be performed.

For the counting of AR-status (sAR(+), sAR(−), and mix) within and adjacent to responding and non-responding factors, only areas around factors which spots contained PCa in combination with >10% epithelial cells, or, stromal spots annotated to 'PCa infiltration in desmoplastic stroma' were included pre-ADT, while post-ADT, all areas around the cancer factor spots were included (Supplementary Fig. 19). The rationale for this is the higher extent of PCa infiltration in stromal areas post-ADT than pre-ADT.

For the DGE analysis, conducted on sAR(+) vs. sAR(−) spots adjacent to PCa-areas (independent of responsiveness), only pure stromal spots were evaluated for selection. Spots in tissue areas that did not clearly have the same tissue structure on the HE- and AR-images were excluded from the analysis. Spots were annotated sAR(−) if not containing any nuclear AR and sAR(+) if more than half of the nuclei were AR-positive. No spots located more than 50% outside the tissue section border was included. Further, to increase the chance of correct selections, only spots located in a cluster of spots with similar AR-annotation was included (minimum 3 spots).

### PCA analysis

The analysis was performed in R version 3.6.1 using the R-package DESeq2[143] version 1.6.3 on rlog-transformed data, with information on patient- and biopsy number and treatment status.

### Differential gene expression and pathway analysis

The analysis was performed in R version 3.6.1 using the R-package STUtility[144], based on the Seurat approach. Genes with a total unique transcript count value of less than 30 across all tissue sections, genes found in less than 10 spots, and spots containing less than 50 transcripts, were filtered away. Normalization and batch correction were achieved using the variance-stabilizing transformation implemented in the SCTransform function from the Seurat package, with the ´nFeature_RNA´ specified as the batch variable in the "vars.to.regress" option. DEGs for each cluster were determined by FindMarkers with parameters min.pct = 0.25, logfc.threshold = X (see main), and test.use = "wilcox". Genes that met criteria for adjusted $p$-value < 0.05 were used for pathway analysis with the network crosstalk-based method BinoX as implemented at PathwAX.sbc.su.se (version I[145]) against pathways in the KEGG database (release 70[146]).

### Cell cycle gene signature analysis in non responding cell clusters

Spots containing non responding cell clusters pre and post ADT are chosen as described above. Read counts of genes, belonging to the 71 cell cycle gene signature are normalized by the total number of read counts per patient (Supplementary Fig. 25).

### Statistics and reproducibility

When comparisons are made between tissue sections it is of fundamental importance to have adjacent sections. If the distance is too large, small clusters of cell types will not be on both sections and the connection between the protein staining and the RNA expression will be blurred. The same is also true for comparison of different protein levels in two slides. To minimise this problem we decided to not make replicate of adjacent sections of the immunohistochemistry stained slides. Instead we used other biopsies, taken at the same time frame and treated in the same way to develop staining protocols that gave high reproducibility.

### Reporting summary

Further information on research design is available in the Nature Research Reporting Summary linked to this article.

## Data availability

### Sequence raw data

Spatial transcriptome sequencing data from prostate cancer needle biopsies that support the findings of this study have been deposited at the European Genome-Phenome Archive (EGA, www.ebi.ac.uk/ega/), which is hosted by the European Bioinformatics Institute (EBI), under accession number EGAS00001006113. The data are available under Data Use Conditions (DUO) and are limited to non-for-profit use as well as health/medical/biomedical purposes. Access is granted if the above is fulfilled and local institutional review board/ethical review board approvals are provided.

### ST and IHC experiments

Count matrices, high-resolution histological, and immunohistochemistry images are available on Mendeley:

Marklund, Maja (2022), "Prostate needle biopsies pre- and post-ADT: Count matrices, histological-, and Androgen receptor immunohistochemistry images", Mendeley Data, V1, https://doi.org/10.17632/mdt8n2xgf4.1. [data.mendeley.com]. Source data are provided with this paper.

## Code availability

Details of the ST analysis pipeline can be found at https://github.com/jfnavarro/st_pipeline.

The factor analysis software (STD) is available under the GNU General Public License v3 at:[https://github.com/SpatialTranscriptomicsResearch/std-nb/tree/5ed3523].

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

## Acknowledgements

This study was supported by The Swedish Cancer Society (J.L., T.H.), AstraZeneca (T.H., E.S., J.L.), and Swedish Foundation for Strategic Research (J.L., T.H.). We thank the National Genomics Infrastructure (NGI, Sweden), for providing infrastructure support. We thank Jonas Maaskola, Joseph Bergenstråhle, Alma Andersson, and Ludvig Larsson for fruitful discussions and data analysis guidance. We also thank Alma Andersson and Mengxiao He, for their effort in proofreading the manuscript.

## Author contributions

M.M., N.S. and E.B. performed the experiments. M.M., S.F., N.S., E.B. and L.B. analyzed the data. M.M., S.F. and N.S. wrote the manuscript, F.T. provided the prostate biopsies, and A.T. and Y.L. annotated the H&E sections. A.E. and A.L. provided clinical expertise. J.L., E.S. and T.H. edited the manuscript. All authors read and approved the final manuscript.

## Funding

## Competing interests

J.L. and M.M. are scientific consultants to 10× Genomics Inc., which holds IP rights to the barcoding technology. The remaining authors declare no competing interests.

## Additional information

**Peer review information** : *Nature Communications* thanks Shalev Itzkovitz and the other anonymous, reviewer(s) for their contribution to the peer review of this work. Peer reviewer reports are available.

