## [Peer review file · Nature Communications]

REVIEWER COMMENTS

Reviewer #1 (Remarks to the Author): Expert in spatial transcriptomics

Marklund et al. use Spatial Transcriptomics to characterize with high spatial resolution the cellular and regional features that are associated with resistance of prostate cancer to castration treatment. They identify pre-existing resistant cancer clones, which presumably undergo EMT following treatment and migrate into the tumor border stroma. They also characterize distinct features of the tumor-associated stroma and identify a simple histological feature – nuclear stromal AR staining, which correlates with resistance and tumor grading. The authors elegantly combine histological features with spatially-resolved transcriptomics and apply state of the art computational methods. The paper is clearly written and provides novel insights into the molecular features of the cancer resistant state. I have some generally minor comments that should be addressed in a revision:

-The STD computational method seems highly informative but is not sufficiently explained in the text. Since it is the core of the current analysis it must be detailed more both in the Results and Methods section. I am specifically pertaining to the spatial transcriptome decomposition, is this a form of principle component analysis? Please elaborate. Also, the factor analysis includes some parameters that seem arbitrary, e.g. the number of factors, how sensitive are the main paper's conclusions to these parameters. Similar comment goes for the determination of responsiveness (rows 769-811), the authors should comment on robustness of the conclusions to variations in the parameters used for determination of these factors.

-Are there single cell RNAseq datasets of human prostate cancer? If such exist, it would be helpful to provide some unbiased classification of spots based on markers of distinct cell types (e.g. as done in PMID 31932730).

-Similarly, if such relevant scRNAseq datasets exist it would be interesting to perform DGE between responder and non-responder spots over the set of genes that are unique to carcinoma cells. This could help differentiate between the differentially expressed spot genes that are clearly stromal (e.g. IGFBP7, MGP, CD74) and those that are carcinoma.

-Discussion, rows 556-558: The questions of whether the resistance cells pre-existed before treatment is very interesting and the identification of resistant spots before treatment supports the picture of pre-existing clones. The claim, however, that evolutionary pressure cannot act over a time frame of 8 weeks is not sufficiently substantiated, I can see situations where massive proliferation and local niche selection could give rise to a de-novo appearing resistant clone. I would rephrase the sentence or substantiate the claim.

-Related to the above point – are cell cycle genes induced in the resistant clones? This should be commented on (if they do, this supports the model of expansion of the pre-existing clones).

-Row 614 – remove “cellular resolution”, the study does not really reach single cell resolution, I would replace with “high spatial”.

- Scale bars should be added to all microscopy images.
- The authors should briefly explain the two grading systems used (GG and GS).
- Line 770 – “TH110 (Photoshop)” – unclear, please explain.
- Line 801 – “Enough with factor activity” – please provide a number.
- Supplementary Figure 7c – There is a typo in the titles (both sides denoted as “before treatment”).

Shalev Itzkovitz

Reviewer #2 (Remarks to the Author): Expert in prostate cancer genomics

In the manuscript by Marklund, the authors applied spatial transcriptomics technology to multiple core needle biopsies collected pre- and post-ADT from 3 prostate cancer patients that were responder, moderate responder and non-responder to ADT. The authors reported that certain cell populations present before treatment exhibited gene expression profiles that matched those of resistant tumor cell clusters, present after treatment. They also observed negative expression of androgen receptor in stroma cells adjacent to resistant clusters.

This is a high quality, interesting study with important findings and significant clinical implications. The pre-existing resistant cells continue to proliferate and spread during the course of treatment, which may explain why some patients become rapidly resistant to ADT.

I have the following concerns:

- 1) It is not clear if the non-responding factors are Neuroendocrine (NE) cells or CRPC-adeno like cells, since the responding and non-responding factors (cells) were defined using nuclear AR activity. NE cells have been constantly found in primary PCa. It would be important to add NE markers, in addition to nuclear AR activity, to determine the nature of the pre-existing therapy resistant cells in primary PCa.
- 2) Although authors analyzed over 4000 barcoded spots from 48 core needle biopsy sections, all spots were collected from 3 primary PCa sample. The pre-existing resistant cells were defined as cells that shared similar profiles with the factors in post-ADT samples. The question is that if the gene profiles of post-ADT cells are proved to be the profiles of resistance? It would be desirable if the authors add CRPC samples into this study, and compare the factors of pre-/post-ADT with factors in CRPC samples.

Alternatively, the authors could test if the signature of post-ADT factors shows significant differences between primary PCa and CRPC samples, using publically available datasets.

3) It will be very helpful for readers to understand those factors in pre- and post ADT samples, if authors could measure the expression of well-known CRPC markers, the oncogenes associated with PCa progression, or CRPC signatures to characterize differences between responding and non-responding factors.

4) Since each spot captures 10–50 cells, each spot contains different types of cells. The authors also reported that those spots can be classified as stroma, 1-10% epithelium, 11-50% epithelium, and 51-100% epithelium. Therefore, the gene profile of each spot can be largely influenced by the percentage of non-epithelial cells within the spot. On Fig. 4a, all differentially expressed genes between responding and non-responding tissue regions in patient 1 are immune related genes, suggesting the difference were confounded by the composition of cells within those regions.

5) The author did not provide the detail procedure of how to adjust the effect of non-epithelial cells on factor calculation. On page 10 line 347, the authors wrote: “The top differentially expressed gene.....average logFC > 0.3 for patient 1 and > 0.5 for patient 2)”. Why were different cut-offs used for patient 1 and patient 2? What is the statistical basis for this selection?

6) Since patient 1 did not show many factors, while patient 3 is a non-responding case, the differential gene expression and pathway enrichment analyses were performed using cell spots only from patient 2, this substantially limited the capacity of delineating the transcriptome differences between non-responding and responding cells. Additional partial response cases may help if feasible.

7) Can authors provide a list of potential markers of the pre-existing resistant cells?

Reviewer #3 (Remarks to the Author): Expert in prostate cancer genomics

This manuscript describes the application of spatial transcriptomics technology to associate gene expression with the response of prostate cancer to androgen deprivation therapy (ADT). Using factor analysis of tumor and stromal cell groups on needle core biopsies, the authors find that the molecular heterogeneity of the samples in the nonresponding patient is highest compared to two other patients with intermediate and favorable response. They also note that nuclear expression of the androgen receptor in epithelial tumor cells as well as stromal cells is associated with response to ADT. The presence of nonresponsive factors in pre-ADT biopsies leads the authors to conclude that therapy-resistant clones are present at the time of therapy initiation and not the result of selective advantage under the therapy pressure.

The study has been performed thoroughly, and the data are presented well although quite hard to follow: The main points could not be understood without following all supplementary data in great detail. Most arguments are based on gene expression signatures (GO classes) and their mutual fractions

in tumor/stromal cells, which is considered as circumstantial experimental evidence only (I am mainly referring to the paragraph "Processes modified ..." lines 331-363 and 380-444).

The factor analyses show that different factors often exhibit considerable similarities in genes. This is also evident from the topological trees (Suppl. Fig. 13-15). The question arises whether the number of factors associated with the various responses might be overestimated, which would consequently lead to different conclusions than the one illustrated in Suppl. Fig. 12. The authors should show convincingly that the factors used are independent from one another.

Despite the large number of biopsies and spatial RNA analyses – the number of patients is small (N = 3). The conclusions might be stronger and the focus of the paper be highlighted if it were possible to validate the existing evidence using more samples (e.g. by determining nuclear AR expression on tissue microarrays) with response data from a large number of patients. Adding such clinical evidence would considerably increase the impact of the findings and convince more readers of the huge potential of this spatial transcriptomics technology.

Co-targeting of epithelial and stromal cells (Discussion, lines 606-612) is a well-known concept in tumor therapy. If the authors have certain suggestions, which are based on the data, I suggest to be more specific here. Otherwise, this section does not add much novelty to the manuscript.

Reviewer #4 (Remarks to the Author): Expert in tumour microenvironment

The authors present data from spatial transcriptomics of prostate cancer. Data is collected from needle biopsies taken pre and post treatment to determine the molecular mechanisms underlying treatment resistant prostate cancer following androgen deprivation therapy. With this they also examine intratumoral heterogeneity

Analysis of transcriptomes identified populations characterized by their gene expression profiles. Unique signatures were indicative of responding and non-responding tumors, irrespective of Gleason scores. Their data also indicated stromal cells adjacent to resistant clusters do not express the androgen and identified differentially expressed genes for these cells. Altogether,

The approach is technologically advanced. While using spatial transcriptomics, the spatial aspect and ultimately the consequences of the location of differentially expressed genes was less well described within the text. To me, this negates the use of the technique in this manuscript over scRNA seq. Why is the climatization of the DE genes within the tissue important? Is this different between pre and post treatment? Does location help define likelihood of therapy response? None of these are covered within the text. From a biologist perspective I found the terminology quite difficult to follow.

- Throughout, there is no protein verification of key differentially regulated genes. This needs to be done, and could easily be done as authors already have sectioned material.
- As a biologist, is it not clear what is meant by a factor, or what a biologically meaningful factor represents.
- Needs markers to distinguish between epithelial and stromal compartments. Not sufficient to rely on nucleus shape or AR expression given the effects treatment has on its expression
- Figure 3: for patient 2, I assume from the RHS panels that the yellow stroma PCA cluster represents pre treatment biopsies compared with chronic inflammation post treatment? This is not clearly annotated.
- Figure 3: For Patient 3, where is the per-treatment stroma cluster given the focus on patients 2 and 3 later?
- Discussion of DE genes, their role and potential functional implications in relation to this study and treatment response is rather superficial.
- Representative Immunostaining of AR in stromal cells for the three patients is not summarized in Supplementary Fig. 13–15. I am not sure which figure the authors are referring to.
- Figure 5: genes identified in sAR- regions pre-treatment such as COL1A1, COL1A2, COL3A1, POSTN, SPARC also relate to collagen deposition and remodelling rather than just EMT. What could the relevance of this be in term of drug penetration rather than resistance? Indeed, figure 5 suggests that increased ECM components is present in areas close to non responding clones pre treatment. Tissue staining of key changing molecules in responders and non responders is needed verify observation, provide insight of tissue morphology and biology.
- Fig 5C: Why did the authors chose to show TIMP and AEBP1 over other genes that look more differentially regulated in the heatmap e.g. col1a1 (AR-) vs. ACTG2 (AR+)?
- Where are the genes such as the key ECM componets located? And how does this change post treatment in responders vs. non responders?

- Also, as highlighted above, markers in addition to AR expression are needed to reliably identify stroma vs. tumour cells. This is the case for all images annotated for stroma and tumour cells

- AR+ stromal cells express higher levels of genes associated with activated myofibroblast phenotype in both patient 2 and 3. What is the biological significance of this?

General points:

- Supplementary figures to not run in order of appearance in text – makes it more difficult to follow
- No scale bars throughout

Dear Editor,

We would like to thank you for your consideration of our manuscript NCOMMS-21-15868A entitled "Spatio-temporal analysis of prostate tumors in situ suggests pre-existence of treatment-resistant clones", and the reviewers for their valuable comments and suggestions. The manuscript was revised according to the reviewers' comments. Please find below our point-by-point responses to the raised issues."

Reviewer 1 - Expert in spatial transcriptomics

Marklund et al. use Spatial Transcriptomics to characterize with high spatial resolution the cellular and regional features that are associated with resistance of prostate cancer to castration treatment. They identify pre-existing resistant cancer clones, which presumably undergo EMT following treatment and migrate into the tumor border stroma. They also characterize distinct features of the tumor-associated stroma and identify a simple histological feature – nuclear stromal AR staining, which correlates with resistance and tumor grading. The authors elegantly combine histological features with spatially-resolved transcriptomics and apply state of the art computational methods. The paper is clearly written and provides novel insights into the molecular features of the cancer resistant state. I have some generally minor comments that should be addressed in a revision:

R1 | Comment 1a

1a: The STD computational method seems highly informative but is not sufficiently explained in the text. Since it is the core of the current analysis it must be detailed more both in the Results and Methods section. I am specifically pertaining to the spatial transcriptome decomposition, is this a form of principle component analysis? Please elaborate.

Answer:

1a: Thanks for this comment, we agree that there was a lack of clarity from our side regarding the STD. In short, each barcoded spot is capturing expression data from several cells simultaneously. STD is trying to unmix the expression from different cells. In that way one obtains the information about what 'cell types' or cell states' (factors) that are present in each spot. Then, a dimensionality reduction of preferred choice is done, in our case UMAP and tSNE. But the classic PCA, or other choices would also work similarly.

We have revised the following sections (yellow highlight):

-Main (page 3, line 126-128)

118 We used a model-based probabilistic framework, Spatial Transcriptome
119 Decomposition⁵² (STD), to perform a data-driven analysis of the gene expression
120 data, which allows for hidden mixture interpretation represented by the non-
121 homogenous cell type composition of the spots. In brief, STD decomposes the spatial
122 gene expression into patterns across the tissue sections, hereby referred to as factors,
123 each representing a distinct gene expression profile with a corresponding spatial
124 activity map. Each factor can be approximated to a specific cell type, cell state,
125 microenvironment, or tissue component, representing different histological
126 conditions. Then, a dimensionality reduction of preferred choice can be done, e.g.,
127 UMAP or tSNE, to visualize the full repertoire of factors in a single image. The full
128 model is described elsewhere⁵³.

-Results (page 8, line 265-269)

260 **Spatially resolved transcriptomes of patient biopsies**
261 Next, we investigated the spatially resolved transcriptomes for the study cases. In
262 total, we generated spatial and transcriptome-wide data for more than 4 000
263 barcoded spots from 48 core needle biopsy sections including two consecutive tissue
264 sections per biopsy. ST data from patients were analyzed individually by applying
265 Spatial Transcriptome Decomposition (STD)⁵² (Extended data files 1–6; schematically
266 overviewed in Supplementary Fig. 2). STD is a probabilistic model that factorizes the
267 observed transcript data into latent gene expression factors (Methods). The factors
268 characterize distinct metagenes, groups of genes that are likely to be co-expressed,
269 and their spatial expression patterns.

-Methods (page 23, line 792-800, 804-805)

792 The factor analysis uses Bayesian shrinkage to avoid overfitting the expression
793 factors. Notably, when extraneous factors are included, their inferred baseline
794 expression levels will be very low. Thus, extraneous factors do not worsen model fit
795 but may make results less interpretable by, for example, introducing noise in
796 visualizations. To accommodate for this fact, we initially overspecified the number
797 of expression factors and then reran the analysis with the number of factors
798 appropriate for our data. This approach avoids underfitting while maximizing the
799 expressiveness and interpretability of the final model. We decided to include factors
800 which had >5,000 transcripts contributing to each factor. 5,000 transcripts are
801 estimated to equal 50-500 cells, which is thus the minimal number of cells we
802 determined to qualify for this analysis.
803
804 The factor analysis software is available under the GNU General Public License v3 at
805 <https://github.com/maaskola/spatial-transcriptome-deconvolution>.

R1 | Comment 1b

1b: Also, the factor analysis includes some parameters that seem arbitrary, e.g. the number of factors, how sensitive are the main paper's conclusions to these parameters.

Answer:

1b: This is indeed an important question and we have hopefully added clarity below (also added in Comment 1a since STD-related):

-Methods (page 23, line 792-800)

792 The factor analysis uses Bayesian shrinkage to avoid overfitting the expression
793 factors. Notably, when extraneous factors are included, their inferred baseline
794 expression levels will be very low. Thus, extraneous factors do not worsen model fit
795 but may make results less interpretable by, for example, introducing noise in
796 visualizations. To accommodate for this fact, we initially overspecified the number
797 of expression factors and then reran the analysis with the number of factors
798 appropriate for our data. This approach avoids underfitting while maximizing the
799 expressiveness and interpretability of the final model. We decided to include factors
800 which had >5,000 transcripts contributing to each factor. 5,000 transcripts are
801 estimated to equal 50-500 cells, which is thus the minimal number of cells we
802 determined to qualify for this analysis.
803

R1 | Comment 2

Similar comment goes for the determination of responsiveness (rows 769-811), the authors should comment on robustness of the conclusions to variations in the parameters used for determination of these factors.

Answer:

We agree with the reviewer that the cutoffs for the parameters should be better explained. It is much appreciated that these details are brought to our attention. We have added the following:

-Methods (page 24, line 833-846)

831 Then, to construct a strict categorization of responsiveness of the tumor factors for
832 upcoming analyses performed, the below criteria were made as a separate
833 categorization of the factors. The rationale for the criteria was that, since we are only
834 looking at small parts of the entire prostate of each patient, we cannot be sure that
835 the biopsies are representative for the whole prostate, and if the patients do really
836 have the, by us, assigned treatment response (on the molecular level). Therefore, to
837 determine what factors were to be considered responding and non-responding, we
838 decided to include several requirements to be met, to increase the chance of
839 including only true responsive factors (or mainly responsive ones) as well as non-
840 responding factors, and to minimize the risk of including tissue spots belonging to
841 non-clear areas regarding responsiveness.
842
843 *Criteria for determination of responsiveness vs non-responsiveness*
844 ● Cutoff of factor activity intensity at a threshold of 110, corresponding to a factor
845 activity of 50% . This was used to exclude spots with relatively low activity of a given
846 factor.
847 ● A minimum of 10 spots per factor
848 ● Requirement of pre-ADT presence of 30-100% spots annotated to cancer. These spots
849 can be either annotated as >10-50% epithelial spots or as stroma with PCa
850 infiltration.
851 ● For the spot-based DGE analysis on factors, described in Fig. 4, spots belonging to
852 both responding and non-responding factors were found in minority, and discarded.
853 Further, if several neighboring spots belonging to a specific factor are annotated to
854 cancer, also neighboring minority non-cancer annotated spots were selected as well if
855 belonging to the same factor.
856

R1 | Comment 3

Are there single cell RNAseq datasets of human prostate cancer? If such exist, it would be helpful to provide some unbiased classification of spots based on markers of distinct cell types (e.g. as done in PMID 31932730).

Answer:

While we agree that this would be helpful, unfortunately we are not aware of any single cell RNAseq dataset of human prostate cancer with sufficiently detailed annotation to permit such an analysis. In order to deconvolve the cell mixture of spots in spatial data similar to the highlighted pancreatic cancer paper, we would need a dataset containing clear cancer versus normal annotations. Compared to some other cancer types it is difficult to separate malignant vs normal prostate cells in single cell analysis. As can be seen in the following studies; PMID

33420488, 32807988 and 33328604, malignant cells are not annotated unfortunately. Indeed, Karthaus's study published in Science, PMID 32355025 generated scRNAseq data from patients with primary prostate cancers. In their work, they attempted to distinguish benign cells from tumor, and were unable to do so using scRNAseq alone.

Further, we found another paper (from Jan 2022) where they, using sc-analysis, managed to sort and identify resistant-like cell types. We did an attempt to use this data and compare it with ours (described in R2 - Comment 2). We could successfully identify cancer epithelial areas. However, their data was limited by few cells and we could not differentiate our responding and non-responding areas using their data.

R1 | Comment 4

Similarly, if such relevant scRNAseq datasets exist it would be interesting to perform DGE between responder and non-responder spots over the set of genes that are unique to carcinoma cells. This could help differentiate between the differentially expressed spot genes that are clearly stromal (e.g. IGFBP7, MGP, CD74) and those that are carcinoma.

Answer:

We agree with the reviewer that by including all genes per spot, we will not obtain a pure repertoire from one and only one cell type, since often the stromal and epithelial cancer cells are mixed. Your suggested idea is good and we would like to try it when possible. However, as our answer in Comment 3, these data sets are not available as of now as per our knowledge. On the positive side however, our results indicate that many of the genes that pop out in our DE-genes belong to the intended cell type. And we reason that the mix of cancer cells with its surrounding stromal and immune related cells might reveal important signatures as well and that the coexistence of these is of high importance.

R1 | Comment 5

Discussion, rows 556-558: The questions of whether the resistance cells pre-existed before treatment is very interesting and the identification of resistant spots before treatment supports the picture of pre-existing clones. The claim, however, that evolutionary pressure cannot act over a time frame of 8 weeks is not sufficiently substantiated, I can see situations where massive proliferation and local niche selection could give rise to a de-novo appearing resistant clone. I would rephrase the sentence or substantiate the claim.

Answer:

We realize that we have not been clear in this statement, and we thank the reviewer for notifying us so we could revise the text in Discussion. The patients in this study have been

treated with a GnRH agonist to decrease the testosterone levels. For this type of drug it takes around 4 weeks to reach castrate testosterone levels. In the first week of the treatment the testosterone levels even increase. In a previous study, 108 of 111 patients had reached castrate testosterone levels after 28 days of treatment (Fujino M et al, *Biochem Biophys Res Commun.* 1974;60:406–413). Normally a treatment period of 6 weeks is considered necessary to assume that the patients have reached castrate testosterone levels. Thus, the actual time of evolutionary pressure due to low testosterone levels is actually around 2 to 4 weeks, and not 8, which makes a major evolutionary change less likely. We have added the following text:

-Discussion (page 19, line 613-617)

613 The rationale of the exclusion of evolutionary pressure is based on the time frame of
614 eight weeks of ADT with a GnRH agonist, which much likely is too short for the
615 presentation of evolutionary pressure, considering that it takes around 4 weeks until
616 castration level of testosterone is reached with this type of ADT. Therefore, the
617 actual androgen depletion time is around 4 weeks per patient, and not 8.

R1 | Comment 6

Related to the above point – are cell cycle genes induced in the resistant clones? This should be commented on (if they do, this supports the model of expansion of the pre-existing clones).

Answer:

We agree with the reviewer that it would be interesting to test for this. We calculated the sum of counts per spot of 71 genes listed as cell cycle related genes that are essential in all cell lines (PMC6927170, additional file 5, Table S4). We observe that cell cycle genes go down due to treatment in the 2 patients, but not in the fully resistant patient 3, who has a lower expression level before treatment. Although the level of cell proliferation and mesenchymal-like state seem to be negatively associated, it might be too early to say to which extent the non-responding cells leave the EMT state and start proliferating in the prostate or at distant sites later. We have added a comment on this in Results and a figure to the supplementary:

-Results (page 16, line 492-494)

492 We also observe that these resistant tumor areas have a lower cell cycle activity as
493 compared to non-resistant areas before treatment onset, when comparing to a 71 cell
494 cycle gene signature¹¹⁴ (Methods, Supplementary Fig. 24).

-Supplementary Fig. 24 (page 33, line 424-427)

424
425
426
427
428

Supplementary Fig. 24: Comparison of the summed read counts of the 71 cell cycle gene signature (Viner-Breuer et al., 2019) in non-responding cell clusters pre- and post- ADT. Each data point represents a spot containing responsive or non-responsive cell clusters either pre- or post-ADT.

R1 | Comment 7-12

Comments and Answers:

-Row 614 – remove “cellular resolution”, the study does not really reach single cell resolution, I would replace with “high spatial”.

We agree with the reviewer that ‘high spatial’ is a better choice, and have revised the text, thanks!

-Discussion (page 20, line 678)

678 In conclusion, this study is the first to present a **high spatial** whole transcriptomic

-Scale bars should be added to all microscopy images.

Thanks for notifying us about this incompleteness in our microscopy images. We have added this.

Example image below:

(Scale bar 50 μm)

-The authors should briefly explain the two grading systems used (GG and GS).

We apologize for being unclear in our explanation of the Gleason grading system. These are not two grading systems, just one. Gleason Grade Group (ISUP 2014) is the new system for Gleason grading, complementing “Gleason Score”, and ISUP has issued a mandate that it should now be used in all publications. We have revised the main text to make this clear:

-Main (page 2, line 57-60)

57 biopsies of the prostate obtained under ultrasound guidance. This system stratifies
58 PCa into different Gleason Scores, ranging from 5 to 10, which further is divided into
59 a simpler form called Gleason Grade Groups (GGs, ISUP 2014), ranging from one to
60 five, where a higher score is associated with a worse outcome in both cases⁵. While)

-Line 770 – “TH110 (Photoshop)” – unclear, please explain.

Thanks for notifying us of this unclear explanation by us. It is now changed in the text:

-Discussion (page 24, line 844-846)

843 *Criteria for determination of responsiveness vs non-responsiveness*
844 • Cutoff of factor activity intensity at a threshold of 110, corresponding to a factor
845 activity of 50% . This was used to exclude spots with relatively low activity of a given
846 factor.

-Line 801 (page 26)– “Enough with factor activity” – please provide a number.

We agree that this was unclearly explained. We have changed this into the following, and hope it will be clearer now:

-Methods (page 25, line 875-876)

874 • Number of spots of the factor post-ADT should be >15%.
875 • A factor was included as a non-responding factor if the factor had activity in a
876 minimum of 10 spots in at least one biopsy post-ADT.

-Supplementary Figure 7c – There is a typo in the titles (both sides denoted as “before treatment”).

Thanks for notifying us of this typo, we have corrected it.

Reviewer 2 - Expert in prostate cancer genomics

In the manuscript by Marklund, the authors applied spatial transcriptomics technology to multiple core needle biopsies collected pre- and post-ADT from 3 prostate cancer patients that were responder, moderate responder and non-responder to ADT. The authors reported that certain cell populations present before treatment exhibited gene expression profiles that matched those of resistant tumor cell clusters, present after treatment. They also observed negative expression of androgen receptor in stroma cells adjacent to resistant clusters. This is a high quality, interesting study with important findings and significant clinical implications. The pre-existing resistant cells continue to proliferate and spread during the course of treatment, which may explain why some patients become rapidly resistant to ADT. I have the following concerns:

R2 | Comment 1

It is not clear if the non-responding factors are Neuroendocrine (NE) cells or CRPC-adenolike cells, since the responding and non-responding factors (cells) were defined using nuclear AR activity. NE cells have been constantly found in primary PCa. It would be important to add NE markers, in addition to nuclear AR activity, to determine the nature of the pre-existing therapy resistant cells in primary PCa.

Answer:

We agree with the reviewer that it is of importance to investigate if the cells in spots with non-responding factors are NE cells or CRPC-adeno-like cells. To address this question we have stained biopsies from patient 2 with the NE marker chromogranin A (CgA). The ratio of CgA positive cells in areas corresponding to areas high in non responding factors in the ST sections was measured. The criteria for the area depicted was that at least five spots with a given factor should be clustered together in both of the ST replicates. The results show that there was no higher ratio of NE cells in areas high in non-responding factors than in other areas of the biopsies. Thus it seems as if the non-responding factors mainly consist of CRPC-adeno-like cells. These results are included in the paper:

-Results (page 11, line 343-365)

343 Neuroendocrine differentiation (NED) can occur in prostate cancer. Prostatic
 344 adenocarcinomas that have undergone NED are resistant to ADT. To investigate if
 345 there was an enrichment of cells that had undergone NED in the areas expressing
 346 resistant factors, we stained all biopsies in patient 2, before and after ADT, with the
 347 neuroendocrine marker chromogranin A (CgA). Areas expressing a given factor
 348 were mapped against the corresponding area in the CgA stained section, and the
 349 ratio of CgA positive cells for the areas was quantified. The criteria for the area
 350 depicted was that at least five spots with a given factor should be clustered together
 351 in both ST-replicates. No difference in ratio of CgA positive cells was found in areas
 352 expressing resistant factors compared to areas expressing non-resistant factors.
 353 (Supplementary Fig. 18-19, Supplementary Table 3-4).
 354 To interpret the spatial RNA data obtained from the ST method it is of importance to
 355 know how well it correlates with the corresponding protein levels in the tissue. In a
 356 large study, including expression data from 60 genes in several different tissues and
 357 cell-lines, the correlation between the number of RNA transcripts and number of
 358 protein molecules was good within each gene, but less accurate when comparing the
 359 number of transcripts with the number of protein molecules in-between different
 360 genes⁶¹. In accordance with this result, we showed in a previous article, that there is
 361 a good concordance between RNA expression detected with ST technology and
 362 protein levels detected with immunocytochemistry for all 7 proteins tested in
 363 prostate tissue applying similar ST protocol used herein⁵⁵. Here we show that this

-Supplementary table 3-4 (page 2, line 15-23)

15 **Supplementary Table 3: The ratio between the chromogranin A and DNA marked area within the**
 16 **region for a given factor. The ratio is expressed as % chromogranin/DNA area.**

Factor	Non-Resistant			Resistant					
	F2	F3	F5	F4	F6	F7	F10	F12	F14
	2.3	3.9	0.1	0.7	1.1	0.0	2.1	0.4	1.9
					0.4	1.2			
					0.0	6.6			
					0.3				
Mean factor	2.3	3.9	0.1	0.7	0.4	2.6	2.1	0.4	1.9
Mean N-R, R	2.1			1.4					

17 Mann-Whitney, resistant factors > non-resistant factors, single sided, p= 0.35

21 **Supplementary Table 4: The ratio between the chromogranin A and DNA marked area for the**
 22 **whole biopsy. The ratio is expressed as % chromogranin area/DNA area.**

Patient 2, Biopsies	2 pre ADT	3 pre ADT	4 pre ADT	1 post ADT	4 post ADT
Mean, whole biopsy	2.6	2.9	1.8	3.0	3.5

23 Mann-whitney, resistant factors (factor values from table 3)>whole biopsies, single sided, p= 0.022

-Supplementary Fig. 18-19 (page 27, line 343-350 and page 28, line 362-369)

343

344 **Supplementary Fig 18. Measuring of cells positive to the neuroendocrine marker chromogranin A.**
345 **a,** A biopsy from patient 2. DNA is depicted in blue and chromogranin A in red. A close up of the
346 white square on the biopsy is shown in the upper mid and named a'. The bar in the figure corresponds
347 to 1 mm and the bars in the close ups correspond to 200 μ m. **b,** The processed image used for
348 measurement of DNA area and chromogranin A area. The borders of the biopsy are eroded by the
349 algorithm (shown in green) to avoid the nonspecific antibody binding that often occur at the tissue
350 rim.

362 **Supplementary Fig. 19. Measuring of the fraction of cells positive to the neuroendocrine marker**
 363 **chromograninA (CgA) in biopsy areas.** Factors shown in green correspond to non-resistant cancer
 364 while factors in purple correspond to resistant cancer. Areas with at least five spots with a given factor
 365 clustered together in the same area in both of the ST replicas were depicted for analyses. The Ratio of
 366 the CgA area and DNA area in these areas was measured and expressed as percentages of CgA area of
 367 the DNA area (Table x) Normally the area of the cytoplasm is approximately three times the area of
 368 the nucleus. Thus, to get an estimation of the percentages of CgA positive cells, the numbers should be
 369 divided by 3. The length of the bars are 1 mm.

R2 | Comment 2

Although authors analyzed over 4000 barcoded spots from 48 core needle biopsy sections, all spots were collected from 3 primary PCa sample. The pre-existing resistant cells were defined as cells that shared similar profiles with the factors in post-ADT samples. The question is that if the gene profiles of post-ADT cells are proved to be the profiles of resistance? It would be desirable if the authors add CRPC samples into this study, and compare the factors of pre-/post-ADT with factors in CRPC samples. Alternatively, the authors could test if the signature of post-ADT factors shows significant differences between primary PCa and CRPC samples, using publically available datasets.

Answer:

We added a publicly available data set (PMID: 20145136), comprising 4 patients diagnosed with androgen-dependent growth (AD) and 4 patients diagnosed with castration-resistant regrowth (CR). We applied the significantly differentially expressed genes between responding and non-responding cell clusters in patient 2 (presented in Fig 4b) to this data set. We found that our gene signature differentiates the PCa patients (AD), from the CRPC patients in this public dataset, see **Review Fig. 1**, suggesting that our post-ADT factors are indeed profiles of resistance.

Review Fig. 1. Gene set from Fig 4b applied to a publicly available data set. The gene set differentiates the AD patients from the CRPC patients into 2 clusters.

Further, we found a newly published paper (PMID: 35058087) where they successfully sorted single cell suspensions from primary prostate cancer, and managed to identify two clusters with resistance-like cells before treatment (X2 and X12). These findings are in line with our findings - that resistant cells can be present before treatment initiation. We performed mapping of these clusters onto our biopsies, using Steroscope (Andersson A., et al., Nature Communications biology, 2020) to see if they matched the areas we had denoted as non-responsive. X12 had no expression in any of our biopsies. However, as shown in **Review Fig. 2**, cluster X2 showed a clear overlap with cancer

epithelial areas in patient 2. Note though that no X2-expression on biopsy 1 is seen, which was the only biopsy of patient 2 that was annotated as non-cancerous by the histopathologist.

When comparing to our responding and non-responding factors, we see that several areas of X2-originated resistance match the factor-based resistant areas. However, we can also see a partial overlap between X2 and our responding factors. This made us want to investigate the data further, and we used Bayesian modeling for inferring cell-type-specific gene expression profiles, see below, and **Review Fig. 3a**.

```
r_tg ~ N(0, sdr)
l_gt ~ N(0, sdl)
X_ng ~ NB(rate = exp(s_n + r_g + r_tg), logit = l_g + l_gt)
```

From this it could be seen that cell type 2 and 12 are uncorrelated, which could indicate insufficient data. To control for this we checked the number of cells in each of the sc-clusters, see **Review Fig. 3b**. X2 has 19 cells and X12 has 13 cells, which is relatively low.

Non-responding factors in patient 2 and responding factors were plotted against cell type 2+12, giving a correlation of 0.23 and 0.61, respectively, see **Review Fig. 3c-d** STD and stereoscope results onto the tissue sections can be seen in **Review Fig. 4-5**. No correlation to our non-responding factors can be seen.

Then, correlations between the cell types and factors were made as seen in **Review Fig. 6**. Cell type 2 and factor 4 have the highest correlation (0.41).

Review Fig. 2: Stereoscope applied on ST sections. a, Overview of all sections. Biopsy sections of patient 2 (biopsy 1-4). **b,** Zoom-in on patient 2 biopsies with corresponding annotations.

Review Fig. 3a-d: Stereoscope applied on ST sections. a, correlations of cell types in the sc-paper. **b**, cell type/cluster number and their corresponding number of contributing cells **c**, Correlation between non-responding cell types X2 and X12 and non-responding STD-data. **d**, Correlation between non-responding cell types X2 and X12 and responding STD-data.

Review Fig 4: Spatial location of summarized non-responding factors compared to stereoscope's non-responding X2 and X12 cell types.

Review Fig 5: Spatial location of summarized responding factors compared to stereoscope's non-responding X2 and X12 cell types.

Review Fig. 6: Correlations between cell types (columns) and factors (rows). Cell type X2 and factor 4 have the highest correlation. However, factor 4 is a responding factor.

To summarize the stereoscope-STD-analysis: We think that the small number of cells in the sc-clusters (13 and 19, respectively) limits the analysis, especially when taking the intra- and intertumorheterogeneity of PCa into account. We also would like to refer to what we state in the paper: “Although patient 3 only displayed non-responding factors, we detected a down-regulation of AR-regulated genes when comparing epithelial spots pre- and post-ADT (Supplementary Fig. 7c). This suggests that spots designated to non-responding factors also contain a fraction of cells that do respond to ADT.” This would mean that some of the cell types within our spots, that the STD-analysis to some extent deconvolves, sometimes get hidden by a stronger gene expression pattern. This would mean that there is a possibility that their resistant-like cell types exist in our spots annotated to responding factors, but since we don’t have them in our STD-data we can’t detect them.

R2 | Comment 3

It will be very helpful for readers to understand those factors in pre-and post ADT samples, if authors could measure the expression of well-known CRPC markers, the oncogenes associated with PCa progression, or CRPC signatures to characterize differences between responding and non-responding factors.

Answer:

We agree that this would be helpful. However, the factors represent gene expression profiles comprising thousands of genes; few selected genes are less powerful to explain fully spatio-temporal changes.

To better explain the concept of a responding and non-responding gene expression profile, the 40 genes of the CRPC gene signature (Gene signature, Jun et al., 2021) were chosen.

In patients 1 and 2, both responders, the CRPC signature genes have a lower mean read count in the non responding spots than in the responding spots. However, comparing the non responding spots pre and post ADT, patient 1 shows a higher fold change than patient 2, the moderate responder. In patient 3, the non-responder patient, the mean read counts of the CRPC signature genes are higher post ADT. See **Review Fig. 7**.

Review Fig. 7: Mean read counts per gene of the CRPC signature.

R2 | Comment 4

Since each spot captures 10–50 cells, each spot contains different types of cells. The authors also reported that those spots can be classified as stroma, 1-10% epithelium, 11-50% epithelium, and 51-100% epithelium. Therefore, the gene profile of each spot can be largely influenced by the percentage of non-epithelial cells within the spot. On Fig. 4a, all differentially expressed genes between responding and non-responding tissue regions in patient 1 are immune related genes, suggesting the difference were confounded by the composition of cells within those regions.

Answer:

Thanks for this comment!

Non-responding DE genes were found in both patient 1 and 2 (patient 3 omitted from the analysis because of lack of identification of responding areas). We hypothesized that these genes, although not all necessarily expressed by carcinoma cells, might be of interest as a group, taking into account the effect of the microenvironment around tumors.

We also reasoned that it is logical to have more differences in the more advanced patient (patient 2) than in the clinical responder (patient 1). This would also go hand in hand with the fact that patient 1 has fewer factors of each category compared with patient 2.

We have made a test to confirm that these genes correlate with non-responding factors and are not solely the result of including all transcripts per spot, as opposed to only genes contributing to each factor. Genes found to be differentially expressed in patient 1 (**Main Fig. 4a**) are upregulated in non-responding factor profile 11 of patient 1.

-Supplementary Fig. 22 (page 31, line 397-402):

396

397

398

399

400

401

402

Supplementary Fig. 22: Genes HLA-DRA, HLA-B, HLA-A and CD74 that were upregulated in spots from areas with non-responding factors in patient 1 were shown to be factor-specific, as opposed to confounding non-factor-specific genes, by comparing the contribution of these genes in all factors of patient 1. It can be seen that these genes are upregulated in non-responding factor 11, relative to the responding areas.

Lastly, we would like to open up for adding these results (or parts of them) into the manuscript if the reviewers think that would be a positive contribution.

R2 | Comment 5

The author did not provide the detailed procedure of how to adjust the effect of non-epithelial cells on factor calculation. On page 10 line 347, the authors wrote: “The top differentially expressed gene.....average logFC > 0.3 for patient 1 and > 0.5 for patient 2”. Why were different cut-offs used for patient 1 and patient 2? What is the statistical basis for this selection?

Answer:

Thanks for this comment, we understand that an added clarity is needed when using different cutoffs as we did here. We used a lower cutoff for ‘Average logFC’ for patient 1 to obtain any DE-genes. This was needed as patient 1 exhibited weaker differential expression. The average logFC for the genes qualifying for our gene heatmap (Average logFC > 0.3 and adjusted p-value < 0.05) was however close to 0.5 in all 4 genes (highlighted in green in Review Figure 1). Two of the four DE genes in patient 1 were also seen in patient 2. We believe it is reasonable that patient 1, that responded better, has fewer DE-genes, while following the same pattern as patient 2.

Review Table 1. Differentially expressed genes for patient 1 responding versus non-responding spots using a cutoff of adjusted p-value and average logFC of 0.05 and 0.3, respectively. The green-marked numbers are close to 0.5.

Gene	Adjusted p-value	Average logFC
HLA-DRA	0.017	-0.445
CD74	0.002	-0.464
HLA-A	0.001	-0.497
HLA-B	0.002	-0.470
HLA-DQB1	0.075	-0.318

R2 | Comment 6

Since patient 1 did not show many factors, while patient 3 is a non-responding case, the differential gene expression and pathway enrichment analyses were performed using cell spots only from patient 2, this substantially limited the capacity of delineating the transcriptome differences between non-responding and responding cells. Additional partial response cases may help if feasible.

Answer:

This is a very important comment from the reviewer, which we are aware of, and have tried to solve. We hoped to get clearer partial responses but unfortunately in the material available we do not have that and if we were about to include more areas of different responses that do not qualify for our relatively harsh criteria, we are afraid of

the risk of losing the separating factors between the 2 conditions. Optimally, we would perform the same experiments on more patients. Unfortunately however, it is not possible for us to obtain more material, otherwise we would be thrilled to do this. This is something that would be great for future continuation.

R2 | Comment 7

Can authors provide a list of potential markers of the pre-existing resistant cells?

Answer:

Thanks for this comment, we agree that providing a list of genes indicative for being elevated in pre-existent resistant PCa cells and also in its surrounding stroma is relevant. Supplementary contains this information now, see below:

-Supplementary Table 5-7 (page 3-5, line 54-77)

54
55
56

Supplementary Table 5: Gene markers up- and downregulated in pre-existent resistant cells (left) in patient 1 and 2 from top 50 genes. Adjusted *p*-value < 0.05 and average logFC > 0.3 for patient 1 and > 0.5 for patient 2). Related to Fig. 4.

Gene markers upregulated in pre-existent resistant cells	Gene markers downregulated in pre-existent resistant cells
DHCR24	SPINK1
SNHG25	NCAPD3
TRPM8	ABHD2
IFI6	TMEFF2
H2AFJ	COL5A2
FKBP2	FABP5P3
EDF1	AGR2
PPDF	HSPA8
CTSD	CTD-2290C23.1
C4B	COX6C
TAGLN	CD46
MYL9	CTD-2287O16.1
APOE	AC016712.2
NBL1	AC010468.1
TIMP1	FABPC1
AEBP1	FABPC3
CD74	SFTPA2
HLA-DRA	GDF15
IGFBP7	MIPEP
MGP	SEMA3C
A2M	MAOA
	EEF1A1P5
	EEF1A1P13
	C1QTNF3
	HPGD
	ACAD8
	CYP2U1
	FAM3B
	SCD

57
58
59
60
61

Supplementary Table 6: Subset of genes upregulated in non-responding spots in patient 2.

Gene	Cancer correlation
DHCR24, TRPM8, IF16	DHCR24 is involved in cholesterol biosynthesis and regulated by the AR, and participates in the conversion of adrenal androgen into its more potent forms; testosterone and dihydrotestosterone. Blocking these pathways might help to reduce treatment resistance (Bonaccorsi et al., 2008, Neuwirt et al., 2020). The androgen regulated TRPM8 , is androgen-regulated and might play an important role in proliferation and apoptosis (Asuthkar et al. 2015), and has further been associated with various cancers (Bidaux et al., 2005, Zhang et al., 2006). IFI6 is an antiapoptotic protein that promotes metastasis in breast cancer (Cheriyath et al., 2018).
CD74	The membrane receptor CD74 is upregulated in the non-responsive areas both in patient 1 and 2, and promotes an increased proliferation, migration and metastatic potential in several cancers (Schröder et al., 2016). It has a role in NF- κ B activation, leading to an immunosuppressive environment. It also plays a role in chemokine production of the potent CCL2 , enabling attraction of myeloid-derived suppressor cells and TAMs to the tumor (Wilkinson et al., 2015). CD74 interacts with the proinflammatory cytokine macrophage migration inhibitory factor (MIF) and has shown to induce EMT (Funamizu et al., 2013).
TIMP1	A secreted glycoprotein that has been linked to promotion of tumor progression by inhibition of apoptosis and stimulation of prostate cancer cell growth (Gong et al., 2013). Elevated levels of TIMP1 levels in plasma predict worse survival outcome in metastatic CRPC patients (Vargas et al., 2008).
IGFBP7, MGP	The tumor stroma marker IGFBP7 (Insulin-like growth factor-binding protein 7) regulates the insulin pathway and has been shown to be elevated in invasive prostate neoplasms (Degeorges et al., 1999). MGP expression is known to be upregulated in CAFs in PCa (Micke et al., 2007). Matrix Gla-protein (MGP) belongs to the family of matricellular proteins (MCP) and induces changes in the extra cellular matrix (ECM) crosslinking and is linked to migration (Mertsch et al. 2009, Gerarduzzi C. et al. 2020).
A2M	Enzymatic activity of PSA-A2M is present in the serum of men with advanced prostate cancer, which in turn affect a range of growth factors such as IL-6 , TGF-beta , PDGF , and FGF , leading to tumor microenvironmental changes (Kostova et al., 2018).

75
76

Supplementary Table 7: Stromal gene markers upregulated and downregulated around pre-existent resistant cells in patient 2 and 3. Adjusted *p*-value < 0.05, average logFC > 0.5.

Gene markers elevated in stroma compartment around pre-existent resistant cells	Gene markers downregulated in stroma compartment around pre-existent resistant cells
SPARC	MSMB
COL1A1	ACTG2
COL1A2	DES
COL3A1	MYH11
SPINK1	MYL9
PTGDS	
CCDC3	
POSTN	
AEBP1	
THBS2	
LRRC32	
FTLP3	
ATXN2L	
SPON2	
FAM3B	
BGN	
TIMP1	
SULF1	
COM	
CXCL14	
SFRP4	

77
78
79
80
81

Reviewer 3 - Expert in prostate cancer genomics

This manuscript describes the application of spatial transcriptomics technology to associate gene expression with the response of prostate cancer to androgen deprivation therapy (ADT). Using factor analysis of tumor and stromal cell groups on needle core biopsies, the authors find that the molecular heterogeneity of the samples in the nonresponding patient is highest compared to two other patients with intermediate and favorable response. They also note that nuclear expression of the androgen receptor in epithelial tumor cells as well as stromal cells is associated with response to ADT. The presence of nonresponsive factors in pre-ADT biopsies leads the authors to conclude that therapy-resistant clones are present at the time of therapy initiation and not the result of selective advantage under the therapy pressure.

R3 | Comment 1

The study has been performed thoroughly, and the data are presented well although quite hard to follow: The main points could not be understood without following all supplementary data in great detail.

Answer:

Thanks for this comment, we understand that this is the case and apologize for that. We have been trying to make it as easy to follow during the circumstances which are that with the word and figure limit it was impossible to provide all information in this paper without extensive supplementary information.

R3 | Comment 2

Most arguments are based on gene expression signatures (GO classes) and their mutual fractions in tumor/stromal cells, which is considered as circumstantial experimental evidence only (I am mainly referring to the paragraph "Processes modified ..." lines 331-363 and 380-444).

Answer:

A commonly used and powerful technique to analyze the function of a set of genes is to apply pathway enrichment analysis (PMID:26125594; PMID:30664679). Indeed, these are not experimental evidence but provide useful information on gene function. This way, biological processes statistically enriched by a gene set can be detected. We have here used a state-of-the-art network based pathway enrichment method and analyzed our gene sets for enrichment of KEGG (PMID:30321428) pathways, which gives a clear and interpretable result.

R3 | Comment 3

The factor analyses show that different factors often exhibit considerable similarities in genes. This is also evident from the topological trees (Suppl. Fig. 13-15). The question arises whether the number of factors

associated with the various responses might be overestimated, which would consequently lead to different conclusions than the one illustrated in Suppl. Fig. 12. The authors should show convincingly that the factors used are independent from one another.

Answer:

We thank the reviewer for this point regarding factor independence. To test the independence of the factors per patient, we have now undertaken a paired Wilcoxon test for each factor combination per patient. The null hypothesis states that the observations (score per gene) in two factors are related, i.e. the observations in one factor are a multiple of the observations in the other factor. The majority of factor combinations of a patient are independent (p-value < 0.1). Only 2 factor combinations in patient 1 (Figure below, red border) and 1 factor combination in patient 2 (red border) were not found to be independent. This figure is added in:

- Supplementary Fig. 16 (page 25, line 321-324).

321
322
323
324

Supplementary Fig. 16: Paired Wilcoxon for each possible factor combination of a patient. P-values equal to 0 are marked in black. Except for two factor combinations in patient 1, and 1 factor combination in patient 2, the factors of a patient are independent.

R3 | Comment 4

Despite the large number of biopsies and spatial RNA analyses – the number of patients is small (N = 3). The conclusions might be stronger and the focus of the paper be highlighted if it were possible to validate the existing evidence using more samples (e.g. by determining nuclear AR expression on tissue microarrays) with response data from a large number of patients. Adding such clinical evidence would considerably increase the impact of the findings and convince more readers of the huge potential of this spatial transcriptomics technology.

Answer:

We agree with the reviewer that increasing the number of patients would further strengthen the impact of our findings. Indeed, we would like to validate the clinical response data in a patient cohort with nuclear AR expression in the tissue post-ADT. However, we do not have such a cohort with co-existent spatial transcriptomic data. Instead we are viewing these results as a showcase that ST technology works well on needle biopsies from prostate pre- and post-ADT and suggests that future larger studies would be needed for clinical translation. The reviewer suggests measuring AR expression on tissue microarrays. However, while this would enable nuclear AR expression to be ascertained, the ST expression data would not be available.

R3 | Comment 5

Co-targeting of epithelial and stromal cells (Discussion, lines 606-612) is a well-known concept in tumor therapy. If the authors have certain suggestions, which are based on the data, I suggest to be more specific here. Otherwise, this section does not add much novelty to the manuscript.

Answer:

We thank the reviewer for flagging this and acknowledge that such concepts have been investigated before. We have revised the text to explain more clearly that selective targeting of AR activity in tumor epithelial but retention of AR signaling in the stromal microenvironment is an important therapeutic goal:

Discussion (page 20, line 667-673)

667 To reduce the risk of relapse of PCa, future potential probably lies in
668 a combined treatment of the tumor and its microenvironment. Co-treatment of
669 stroma and tumor is an important concept in cancer therapy. In the case for PCa
670 treatment however, an intrinsic problem with ADT is that the tumor epithelial cells
671 are the desired targets, but the stroma will also be targeted, which will induce a
672 more lethal microenvironment since AR in stroma is required for a healthy
673 phenotype. To overcome this, drugs could be developed targeting only epithelial
674 cells by decreasing their AR signaling, which could be achieved by targeting AR co-
675 regulators and pioneer factors that are specific for only epithelial cells, as suggested
676 earlier¹³⁶.
677

Reviewer 4 - Expert in tumor microenvironment

The authors present data from spatial transcriptomics of prostate cancer. Data is collected from needle biopsies taken pre and post treatment to determine the molecular mechanisms underlying treatment resistant prostate cancer following androgen deprivation therapy. With this they also examine intratumoral heterogeneity

Analysis of transcriptomes identified populations characterized by their gene expression profiles. Unique signatures were indicative of responding and non-responding tumors, irrespective of Gleason scores. Their data also indicated stromal cells adjacent to resistant clusters do not express the androgen and identified differentially expressed genes for these cells. Altogether,

R4 | Comment 1

The approach is technologically advanced. While using spatial transcriptomics, the spatial aspect and ultimately the consequences of the location of differentially expressed genes was less well described within the text. To me, this negates the use of the technique in this manuscript over scRNA seq.

Answer:

We agree with the reviewer that scRNA has its clear advantages and can be simpler and cheaper than spatial transcriptomics. However, in this case, we would like to argue that the importance of ST lies in the fact that, together with the STD (factor) analysis, it can show the spatial locations of cell clusters with similar gene profiles, giving information about whether or not a gene profile is present only pre-ADT (indicative of responding cancer cells), or both pre- and post-ADT (indicative of non-responding cancer cells). This biological question would not be possible to answer with scRNA-seq alone since the spatial integrity is destroyed, by the very nature of the technology. Moreover, with ST it is possible to identify the spatial distribution of the gene expressions across the tissue sections, meaning we can e.g. determine if a factor is sparse and spread out or located at 1 single cluster. Also, the fact that surrounding cells can be annotated and profiled regarding expression contributes to a larger understanding of the tissue function.

Nonetheless, we recognise the potential for scRNA-seq (or IHC) as a simpler, cheaper identification of these genes in a clinical setting, once broader validation of the genes of interest in spatial material from both responders and non-responders has taken place.

If the author prefers, we are open to adding a few sentences into the paper about this.

R4 | Comment 2

Why is the climatization of the DE genes within the tissue important? Is this different between pre and post treatment? Does location help define likelihood of therapy response? None of these are covered within the text.

Answer:

Thank you for raising these interesting questions. We would like to argue that the DE genes found between the different states of responsiveness and its neighboring stroma is important since they could be used clinically (after validation in more patients), to find out whether the patient is at high risk of low response to ADT. We only looked before treatment since post-treatment tissue would be affected by the ADT. The location of the DE genes will not help define the likelihood of therapy response, instead we plotted the expression and location of the DE-genes to validate that they were more prone to be located in the factors we extracted the expression data from (as seen in **Supplementary Fig. 23**).

R4 | Comment 3

From a biologist perspective I found the terminology quite difficult to follow.

Answer:

We appreciate the reviewer's comment that the terminology is quite hard to follow, and we have tried our best to make things clearer, with a main focus on the STD analysis, see below. We would like to refer to the answer we gave to Reviewer 1 Comment 1, in which we have made an effort to clarify what STD really does.

Further, in the review Asp et al., 2020 (<https://doi.org/10.1002/bies.201900221>) the main concepts in spatially resolved transcriptomics are covered in detail and could serve as a deeper explanation to many terms used in this manuscript.

R4 | Comment 4

Throughout, there is no protein verification of key differentially regulated genes. This needs to be done, and could easily be done as authors already have sectioned material.

Answer:

We agree with the reviewer that protein verification of expressed genes of interest is always important. In recognition of this, in our previously study, "Spatial maps of prostate cancer transcriptomes reveal an unexplored landscape of heterogeneity", Nature Communications, 2018, 9, 1–13, we included a comprehensive protein-level validation, for 7 discriminatory genes of interest (Fig. 2 and Supplementary Figs. 13–14).

Having undertaken such validation in our previous experiments with ST, we instead chose to focus on subcellular AR immunostains of neighboring sections given the importance of this

element in the current work. We wish to emphasize that due to degradation of tissue sections during the ST protocol it is not possible to undertake further protein analysis of the identical sections themselves. We have intentionally relied on our previous validations.

However, we have compared the ST expression of AR RNA with AR protein levels from sections relatively close to each other, showing that we have a good correlation between RNA expression and protein levels in this material.

The data has been included in the supplementary as Supplementary Fig. 20 and a paragraph has been added in the article:

-Results (page 11, line 354-365):

“To interpret the spatial RNA data achieved from the ST method it is of importance to know how well it correlates with the corresponding protein levels in the tissue. In a large study, including expression data from 60 genes in several different tissues and cell-lines, the correlation between the number of RNA transcripts and number of protein molecules was good within each gene, but less accurate when comparing the number of transcripts with the number of protein molecules in-between different genes (Edfors et al Molecular Systems Biology 12: 883 | 2016). In accordance with this result, we showed in a previous article, that there is a good concordance between RNA expression detected with ST technology and protein levels detected with immunocytochemistry for all 7 proteins tested in prostate tissue applying the same ST protocol used herein (landscape, Nat Com.). Here we show that this relationship between RNA expression and protein levels also is valid for the androgen receptor (Supplementary Fig. 20).”

R4 | Comment 5

As a biologist, is it not clear what is meant by a factor, or what a biologically meaningful factor represents.

Answer:

We thank the reviewer for highlighting this statement by us. We have considered this point and acknowledge that the phrasing ‘biologically meaningful factors’ is rather poor. Since we cannot be sure where the cutoff between a biologically meaningful and a non-meaningful factor goes, we wanted to pre-determine a set of harsh criteria with the goal to create a good margin between those states, to minimize factors with overlapping traits.

What we did was to choose an arbitrary cutoff including only factors having more than 5000 transcripts, because that amount of transcripts represents a relatively high total number of contributing cells across the tissue sections. In this way we increase the chance of including biologically relevant factors (since they are most likely having a global and rather robust tissue pattern), however at the same time we also increase the risk of neglecting biologically relevant factors having fewer contributing transcripts, e.g. caused by only few unique cancer cells. However this risk is motivated in order to obtain the margin between the responsiveness states mentioned above.

We have added a section in Methods where we hopefully explain this in an improved way (also included in the answer to Reviewer 1 Comment 1b since STD-related):

-Methods (page 23, line 792-800)

792 The factor analysis uses Bayesian shrinkage to avoid overfitting the expression
793 factors. Notably, when extraneous factors are included, their inferred baseline
794 expression levels will be very low. Thus, extraneous factors do not worsen model fit
795 but may make results less interpretable by, for example, introducing noise in
796 visualizations. To accommodate for this fact, we initially overspecified the number
797 of expression factors and then reran the analysis with the number of factors
798 appropriate for our data. This approach avoids underfitting while maximizing the
799 expressiveness and interpretability of the final model. We decided to include factors
800 which had >5,000 transcripts contributing to each factor. 5,000 transcripts are
801 estimated to equal 50-500 cells, which is thus the minimal number of cells we
802 determined to qualify for this analysis.
803

R4 | Comment 6

Needs markers to distinguish between epithelial and stromal compartments. Not sufficient to rely on nucleus shape or AR expression given the effects treatment has on its expression.

Answer:

We appreciate the reviewers comment and agree that markers to distinguish between epithelial and stromal cells would serve as a trustful validation of these cell types. However in this case, we would like to highlight five points that served as rationale for our decision to not include this validation strategy.

- This first point partly overlaps with our answer to ‘Comment 4 by Reviewer 4’: Since the ST-sections are degraded and the closest consecutive tissue sections were used for AR-staining, we don’t have any tissue sections that would, with high similarity, represent the ST-spots.
- We are of the rather strong opinion that an experienced histopathologist is skilled to differentiate between stromal and epithelial cells. Therefore we used urohistopathologists to annotate the ST-sections. To minimize the risk of annotation discrepancies, we asked two urohistopathologists to annotate the tissue sections, the first one for larger areas of cell types and the second one for a spot-wise analysis.
- In addition to these manual annotations, we used the in-house-developed program that identifies nucleus shapes and thus differentiates between stromal and epithelial cells.
- AR-staining was done pre- and post-ADT but as we interpret the reviewer’s comment, which we agree upon, the ADT might induce unknown (confounding) effects on the AR-staining. Therefore, we excluded all AR-stained tissue sections *post-ADT* in the analyses made on epithelial and stromal spots.
- As a separate validation based on the ST-data, to ensure we could separate stromal from epithelial cells was to extract the ST data from spots consisting of (close to) pure

epithelial or stromal cells and performed DE-analysis (see Supplementary Fig. 6b). In this way we could see a substantial difference between the two compartments.

R4 | Comment 7

Figure 3: for patient 2, I assume from the RHS panels that the yellow stroma PCA cluster represents pre-treatment biopsies compared with chronic inflammation post-treatment? This is not clearly annotated.

Answer:

Yes, the reviewer's observation is correct, and we apologize for not being clearer with this in the text. The light yellow UMAP cluster consists mainly of factor 1 i.e. chronic inflammation and stroma from pre-ADT, and the blue UMAP cluster, annotated to the same, is mainly from post-ADT, see **Review Fig. 8**.

Review Fig. 8: Part from Supplementary Fig. 10 (patient 2) shows that Factor 1 has factor activity pre-ADT, as opposed to factor 15 and 16 which present a more sparse pattern, mainly located post-ADT.

R4 | Comment 8

Figure 3: For Patient 3, where is the pre-treatment stroma cluster given the focus on patients 2 and 3 later?

Answer:

The purple-like spots (not within circles in Fig. 3c) correlates to factor 1, see **Review Fig. 9**, and is present both pre- and post ADT. Several other factors have cancer mixed with stroma.

Review Fig. 9: Part from Supplementary Fig. 11 (patient 3) shows that Factor 1 is the only factor that mainly contains stroma and that it is present both pre- and post-ADT.

R4 | Comment 9

Discussion of DE genes, their role and potential functional implications in relation to this study and treatment response is rather superficial.

Answer:

We summarized the findings in the main text, and provide a more detailed discussion of the potential functional implications of the most important DE genes in Supplementary Table 6. We have further applied pathway enrichment of KEGG pathways, and have highlighted the observed enrichment of pathways most relevant to the studied biological conditions. We would also like to refer to our answer to Reviewer 4, Comment 3.

R4 | Comment 10

Representative Immunostaining of AR in stromal cells for the three patients is not summarized in Supplementary Fig. 13–15. I am not sure which figure the authors are referring to.

Answer:

We thank the reviewer for observing this miss from our side. We have revised the text to what we intended to, which was to explain that the proportion of AR activity is illustrated as a pie chart per factor for each of the patients in Supplementary Fig. 13-15. See below for the change we have done:

Results (page 16, line 503-504):

503 The proportions of stromal AR-staining in stromal nuclei is illustrated as a pie chart
504 per factor for each of the patient in Supplementary Fig. 13–15. We hypothesized that
505 a responding tumor factor would have a higher extent of surrounding stromal AR-
506 positive (sAR(+)) cells, while non-responding tumor factors would be encircled by
507 sAR(-) cells.
508

R4 | Comment 11

Figure 5: genes identified in sAR- regions pre-treatment such as COL1A1, COL1A2, COL3A1, POSTN, SPARC also relate to collagen deposition and remodeling rather than just EMT. What could the relevance of this be in terms of drug penetration rather than resistance? Indeed, figure 5 suggests that increased ECM components is present in areas close to non responding clones pre treatment. Tissue staining of key changing molecules in responders and non responders is needed verify observation, provide insight of tissue morphology and biology.

Answer:

We agree with the reviewer that the collagen deposition and remodeling probably also are an ongoing process in the AR-stroma. As the reviewer implicates, this could be of fundamental relevance for drug penetrance when treating prostate cancer with various drugs. A study comparing the drug concentration in tumors surrounded with AR+ or AR- stroma would bring knowledge to this issue and reveal if tumors surrounded by AR- stroma are undertreated and thereby lead to continuous tumor growth or relapses. Clinically, it would be of importance to know if any of the drugs used today for cancer treatment have this penetration problem to be able to avoid using them in patients with AR-stroma. Further, it would give the pharma companies motivation to develop drug analogues with better penetrance properties in AR- stroma.

However, in this study, the treatment given (GnRH agonists) will, after a time delay, decrease the number of GnRH receptors in the pituitary gland, leading to a reduction of the release of luteinizing hormone and thereby blocking the production of testosterone in the testis. Thus, the drug used herein does not target the tumors directly and therefore the permeability properties of the environment around the tumors do not affect the efficacy of the drug.

R4 | Comment 12

Fig 5C: Why did the authors chose to show TIMP and AEBP1 over other genes that look more differentially regulated in the heatmap e.g. col1a1 (AR-) vs. ACTG2 (AR+)?

Answer:

We agree that the reason for this is not very clear. However we chose to show TIMP1 and AEBP1 because these genes were relatively highly differentially expressed in our data and of special interest because of the previous studies showed the importance of elevated levels of TIMP1 and AEBP1 to predict worse survival outcomes in metastatic CRPC patients (Vargas et al., 2008).

R4 | Comment 13

Where are the genes such as the key ECM components located? And how does this change post treatment in responders vs. non responders?

Answer:

The genes listed in Figure 5 are the result of the comparison of AR(+) stroma versus AR(-) stroma cell clusters within and adjacent to responding and non-responding factors. Their significantly higher or lower gene expression in sAR(-) spots was observed pre-ADT in areas in direct vicinity of such regions, whose spots contained PCa in combination with >10% epithelial cells, alternatively stromal spots annotated to 'PCa infiltration in desmoplastic stroma'.

Post-ADT, all areas in direct vicinity of the non-responding cancer factor spots were considered (Method, Supplementary Figure 18).

To spatially relate how this changes pre- and post-treatment, we chose the genes COL1A1 and COL1A2, which are ECM components, and significantly differentially expressed in sAR(-) vs sAR(+) as listed in **Review Fig. 10** (Figure below).

Review Fig. 10: Spatial association of genes COL1A1 and COLA2 to non responding cell clusters pre and post ADT in patient2 (first and second row), moderate responder, and in patient3 (third and fourth row), non responder.

R4 | Comment 14

Also, as highlighted above, markers in addition to AR expression are needed to reliably identify stroma vs. tumour cells. This is the case for all images annotated for stroma and tumour cells.

Answer:

We would like to refer to the answer to Comment 6 by Reviewer 4.

R4 | Comment 15

AR+ stromal cells express higher levels of genes associated with activated myofibroblast phenotype in both patient 2 and 3. What is the biological significance of this?

Answer:

Yes, the AR+ stromal regions express higher levels of the muscle protein coding genes ACTG2, DES, MYH11 and MYL9 which all are associated with activated myofibroblasts. However, they are also associated with smooth muscle cells whose expression of these genes are higher than in myofibroblasts. In contrast to stroma in many other tissues, smooth muscle cells are an abundant cell type in normal prostate stroma.

An increase of expression in these muscle protein coding genes in stroma surrounding tumors is often interpreted as that a transformation of normal fibroblast to myofibroblast has occurred. However, this assumption is only valid in stroma not harboring a large fraction of smooth muscle cells.

Since prostate stroma is rich in smooth muscle cells, we believe that the expressions from these muscle-associated genes mainly originate from smooth muscle cells. In the AR+ stroma the remodeling of the stroma has not gone as far as in the AR- stroma and the amount of normal smooth muscle cells is therefore higher, giving rise to higher expression values of these genes. Further, myofibroblast expresses a higher amount of collagen than normal fibroblast and in the AR- stroma we do see an increased expression of collagen.

The MSMB gene is the only non-muscle associated gene in AR+ stroma that is overexpressed. This gene codes for prostate secretory protein 94. The MSMB gene is downregulated in prostate cancers, and the degree of downregulation is correlated to the aggressiveness of the cancer, which also supports the view that the AR+ stroma is a less remodeled form of stroma than the AR – variant (The Prostate. 2018; 78:257–265).

R4 | Comment 16

- *Supplementary figures to not run in order of appearance in text – makes it more difficult to follow*
- *No scale bars throughout*

Answer:

We appreciate that the reviewer highlights these inconsistencies. We have ordered them as they appear in the text, and added scale bars to IHC-images.

REVIEWERS' COMMENTS

Reviewer #1 (Remarks to the Author):

The authors have satisfactorily addressed of all my points.

Reviewer #2 (Remarks to the Author):

I am happy with the revised manuscript.

Reviewer #3 (Remarks to the Author):

My previous points have been considered appropriately. I have no further comments.

Reviewer #4 (Remarks to the Author):

Thank you to the authors for taking on board the reviewers' comments, and making a substantial effort to address them. I enjoyed reading the revised manuscript and would be happy to see it published.

Comment 1:

The authors misunderstood my meaning here. I think the use of spatial transcriptomics is technologically advanced and they use it well, but I would like to see the authors highlight the advantages that spatial transcriptomics brings over scRNAseq and why they used it. The power of this didn't really come through in the original text. In fact, they argue it really well in their response, and it might be good to include something along these lines somewhere in the discussion.

Minor point:

supplementary fig 16 still appears in the text (p9 line273) before supp figures 9-11 are mentioned (p9 line 276).

Minor point: p17 line 550. Do you mean figure 5d or 5c?

Point-by-point responses to referees - NCOMMS-21-15868A

Please find below our point-by-point responses to the raised issues.

Reviewer 4 - Expert in tumor microenvironment

Thank you to the authors for taking on board the reviewers' comments, and making a substantial effort to address them. I enjoyed reading the revised manuscript and would be happy to see it published.

R4 | Comment 1

The authors misunderstood my meaning here. I think the use of spatial transcriptomics is technologically advanced and they use it well, but I would like to see the authors highlight the advantages that spatial transcriptomics brings over scRNAseq and why they used it. The power of this didn't really come through in the original text. In fact, they argue it really well in their response, and it might be good to include something along these lines somewhere in the discussion.

Answer: Thanks for the clarification. As suggested, we have included part of the arguments in the last paragraph of the Discussion. The updated paragraph is as follows:

"In conclusion, this study presents a high spatial whole transcriptomic analysis of biopsies before and after ADT. We want to highlight that the power of using ST lies in showing the spatial locations of cell clusters with similar gene profiles. The biological question would not be possible to answer with, e.g., single-cell RNA-seq alone, since the very nature of the technology eliminates the spatial integrity. Moreover, with ST, it is possible to identify the spatial distribution of the gene expressions across the tissue sections, meaning we can e.g. determine if a factor is sparse and spread out or located at one single cluster. Further, the neighborhood of tumor cells can be analyzed within the same experiment. Here we identified tumor cells with castration-resistant potential that are present already before treatment, and characterized potential biomarkers that may provide the advantage of being more specific and effective in future clinical management of PCa. We also characterized the gene expression of resistant cells' neighboring nuclear AR-negative stromal cells. Overall, we demonstrate the importance of a combined temporal and spatial analysis of a tumor in the context of its microenvironment suggesting a new course of action to understand treatment resistance."

R4 | Minor points

-Supplementary fig 16 still appears in the text (p9 line273) before supp figures 9-11 are mentioned (p9 line 276).

Answer: Sorry for not changing this after the first round of comments; we, unfortunately, didn't notice this mistake. Now it is changed.

-p17 line 550. Do you mean figure 5d or 5c?

Answer: We meant 5c; thanks for being observant and sorry for the inconvenience. We have now changed this.